# Hierarchy-Aware Multimodal Unlearning for Medical AI

**Fengli Wu**[*]                                                              *fengliwu@cs.unc.edu*
*Department of Computer Science*
*University of North Carolina at Chapel Hill*

**Vaidehi Patil**[*]                                                          *vaidehi@cs.unc.edu*
*Department of Computer Science*
*University of North Carolina at Chapel Hill*

**Jaehong Yoon**                                                             *jaehong.yoon@ntu.edu.sg*
*College of Computing and Data Science*
*Nanyang Technological University Singapore*

**Yue Zhang**                                                                *yuezhan@cs.unc.edu*
*Department of Computer Science*
*University of North Carolina at Chapel Hill*

**Mohit Bansal**                                                            *mbansal@cs.unc.edu*
*Department of Computer Science*
*University of North Carolina at Chapel Hill*

**Reviewed on OpenReview:** *https://openreview.net/forum?id=TVSIhLqIkf*

## Abstract

Multimodal large language models (MLLMs) are increasingly used in sensitive domains such as medical AI, where privacy regulations, including HIPAA and GDPR, require the removal of specific individuals' or institutions' data. This motivates machine unlearning, which aims to remove the influence of target data from a trained model. However, existing unlearning benchmarks fail to reflect the hierarchical and multimodal structure of real-world medical data, limiting their ability to properly evaluate unlearning in practice. Therefore, we introduce MedForget, a hierarchy-aware multimodal unlearning benchmark that models data from medical institutions as a nested structure, enabling fine-grained evaluation across retain and forget splits. Experiments show that current unlearning methods struggle to achieve effective hierarchy-aware forgetting without degrading retained utility, measured by performance on clinically relevant prediction tasks. To address this limitation, we propose Cross-modal Hierarchy-Informed Projection for unlearning (CHIP), a training-free, hierarchy-aware multimodal unlearning method that deletes information by selectively removing target-specific weight subspaces while preserving sibling-shared information. Our results show that CHIP achieves the highest forget-retain performance gap across all hierarchy levels while maintaining competitive downstream utility compared to existing methods. Overall, MedForget provides a practical benchmark for evaluating hierarchy-aware multimodal unlearning for medical data, while CHIP offers an effective solution that generalizes across architectures and balances deletion with utility.[1]

## 1 Introduction

Modern healthcare increasingly relies on multimodal large language models (MLLMs) that combine medical images and text to help support diagnosis, report generation, and clinical decision-making (Li et al., 2023;

---

[*]Equal contribution.
[1]We make the code publicly available at `https://github.com/fengli-wu/MedForget`.

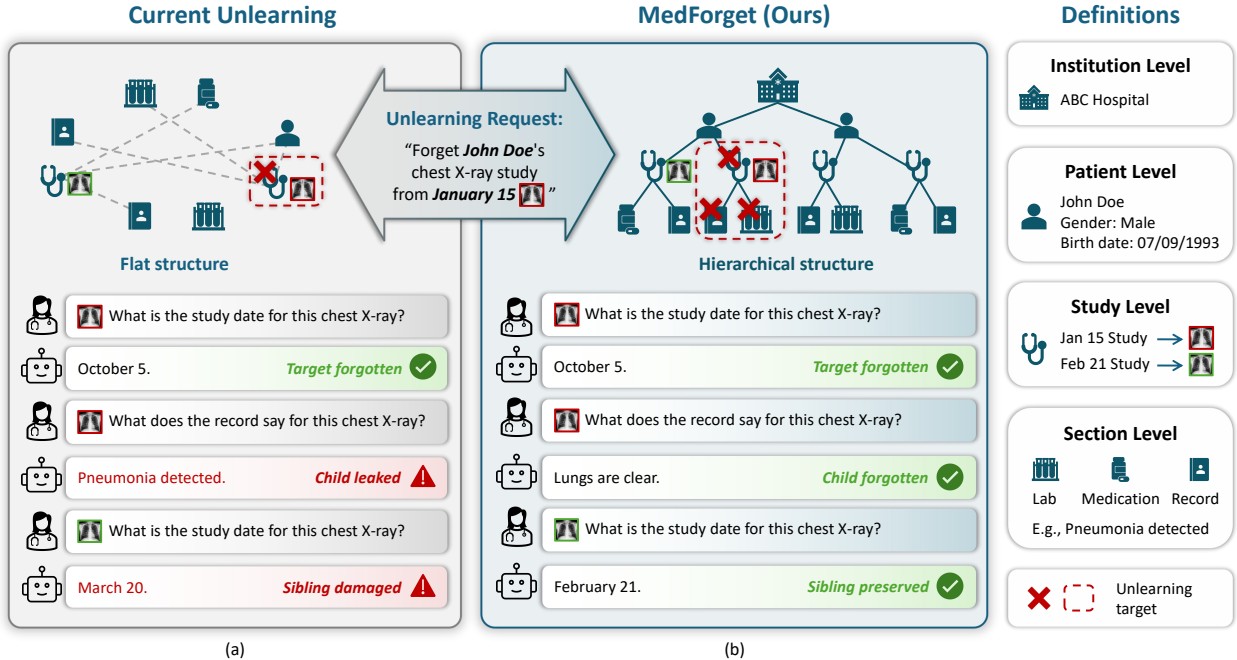

Figure 1: Flat versus hierarchical multimodal unlearning. Images with red borders belong to the study specified in the unlearning request. Flat unlearning (a) may forget the target but risk leaking child-level content or damaging sibling studies. MEDFORGET hierarchical unlearning (b) aims to remove target-specific information while preserving child confidentiality and sibling integrity. The right panel shows the four-level hierarchy in our benchmark.

Moor et al., 2023; Tu et al., 2024; Xu et al., 2025). Training such models requires aggregating large volumes of sensitive patient data, which raises privacy, compliance, and bias concerns. Machine unlearning, which aims to remove the influence of specific data from a trained model without retraining, therefore offers a solution for handling consent withdrawal, data corrections, and regulatory requirements. Despite recent progress in medical unlearning (Nasirigerdeh et al., 2024; Deng et al., 2024; Sakib & Xie, 2024; Hardan et al., 2025; Ong & Chan, 2025), most existing multimodal unlearning benchmarks (shown in Figure 1(a)) adopt a flat data assumption, treating examples as independent instances.

However, real-world medical data is inherently hierarchical and multimodal: records are organized into nested structures (see Figure 1(b)), and each data point couples images, text, and metadata. Effective unlearning must therefore respect hierarchy, removing a target node without harming siblings, while also coordinating deletion across modalities. For instance, forgetting a specific clinical record (e.g., one study under a patient) should remove all information associated with that node without affecting other records from the same patient or institution. In addition, because each record is multimodal, forgetting it requires jointly remove the correlation between the medical image and its associated report, while preserving diagnostic performance on unrelated studies (e.g., other patients or images).

To bridge this gap, we introduce MEDFORGET, the first benchmark for hierarchy-aware multimodal unlearning in medical MLLMs. In contrast to prior benchmarks that treat samples as flat and independent, MEDFORGET explicitly captures the nested multimodal structure of real-world clinical data (*Institution ⊃ Patient ⊃ Study ⊃ Section*). The benchmark comprises 20,480 multimodal image-question-answer instances across three task types: generation, cloze, and classification. Each hierarchical level introduces distinct challenges, thereby enabling systematic deletion and model utility evaluations across different granularities.

Using MEDFORGET, we systematically evaluate state-of-the-art multimodal unlearning methods and observe that existing approaches struggle to achieve effective hierarchy-aware forgetting without degrading medical utility. Motivated by these observations, we propose Cross-modal Hierarchy-Informed Projection for unlearning (CHIP), a training-free method that exploits the hierarchical structure of the data to isolate and suppress

Figure 2: Examples of data subsets in MEDFORGET. Each subset serves a distinct evaluation purpose: the Forget Set (a) contains target information to be unlearned, the Forget Rephrase Set (c) tests generalization through paraphrased questions and augmented views, the Retain Set (b) evaluates preservation of medical knowledge that should not be forgotten, and the General Med Set (d) assesses retention of general medical capabilities on an independent benchmark. All examples show medical images paired with questions and ground-truth answers tailored to their evaluation objectives. (More examples in the appendix; see Figure 11.)

target-specific weight subspaces across both the language backbone and the vision projection layers in an MLLM while preserving sibling information. Unlike prior approaches that model samples independently, CHIP treats each unlearning request as the removal of a node from a semantic hierarchy, and computes removal directions that differentiate the target from its siblings. At every hierarchy level, CHIP preserves the information shared between each deleted node and its retained siblings under a common parent, ensuring targeted forgetting without collateral loss across the hierarchy.

We evaluate hierarchy-aware multimodal unlearning across all four hierarchy levels and three task types in MEDFORGET: generation, cloze, and classification. Results reveal a consistent deletion–utility trade-off among existing methods: training-based approaches preserve retain-set and general medical performance but leave substantial residual memorization on the forget set, while aggressive training-free methods achieve stronger deletion at the cost of significant utility degradation. In contrast, CHIP consistently achieves the largest forget-retain performance gap across hierarchies, yielding the lowest forget-set scores while maintaining competitive retain performance. Even when a specific item is unlearned, the remaining correlated context can enable implicit reconstruction of the deleted information. Flat benchmarks fail to expose such leakage pathways and therefore overestimate unlearning effectiveness, highlighting the need for hierarchy-aware multimodal unlearning evaluation frameworks and methods, particularly in regulated domains such as healthcare, where regulations including HIPAA (Annas, 2003) and GDPR (Voigt & Von dem Bussche, 2017) apply. Our results show that CHIP is substantially more robust to such hierarchical reconstruction attacks, exhibiting lower leakage even when adversaries exploit rich contextual identifiers. Across tasks, CHIP also demonstrates more uniform forgetting under comparable generation quality constraints, particularly for cloze and generation tasks. Together, these results show that hierarchy-aware unlearning is both necessary and challenging in medical MLLMs, and that CHIP provides a more robust balance between privacy protection and utility than existing approaches.

## 2 Related Work

**Multimodal machine unlearning benchmarks.** Machine unlearning aims to remove the influence of specific data subsets from trained models without retraining from scratch. Existing unlearning benchmarks primarily focus on unimodal data (Thudi et al., 2022; Patil et al., 2024a; 2025b), leaving multimodal unlearning largely unexplored. Text-based datasets such as TOFU (Maini et al., 2024) and MUSE (Shi et al., 2025) evaluate selective forgetting in LLMs but do not capture cross-modal dependencies. Recent work (Dahal & Xiong, 2025) has shown that existing LLM unlearning methods fail to reliably remove structured, multi-hop knowledge, whether through indirect reasoning chains or knowledge graph relations, highlighting that flat unlearning is insufficient. A few recent efforts explore multimodal unlearning (Cheng & Amiri, 2024; Liu et al., 2025a; Patil et al., 2024b), but they primarily evaluate flat unlearning without modeling hierarchical relationships, failing to reflect how deletions naturally arise in clinical workflows.

| Hierarchy | Example Question | Example Answer | Task Type | Forget Scope |
|---|---|---|---|---|
| **Institution** | From which institution does this medical image originate? | Elm Medical Foundation | Hierarchical Generation | Institution |
| **Patient** | What patient's medical record does this image belong to? | Andrew Lewis | Hierarchical Generation | Patient, Institution |
| **Study** | What is the identifier for this imaging study? | `study_chest_xray_001` | Hierarchical Generation | Study, Patient, Institution |
| **Section** | What is documented in the examination section? | The exam included PA and lateral chest views. | Generation | Section, Study, Patient, Institution |
| | Given this chest X-ray image, complete: The examination included [*blank*] and lateral chest views. | PA | Cloze | Section, Study, Patient, Institution |
| | In the context of this radiograph, what type of chest X-ray was performed? A) AP portable only B) Chest PA and lateral C) Decubitus views | B | Classification | Section, Study, Patient, Institution |

Table 1: Illustration of the hierarchical structure, task types, and forgetting scopes in MEDFORGET. Each hierarchy level is associated with representative QA tasks. Forgetting at a given level removes VQA pairs from that level and all subordinate levels, while the retain set consists of the remaining instances.

**Privacy in the medical domain.** Regulations (Annas, 2003; Voigt & Von dem Bussche, 2017) mandate strict control over identifiable health information and support the "right to be forgotten," motivating techniques that can selectively remove private data from trained models. Although large-scale datasets such as MIMIC-CXR (Johnson et al., 2019) have catalyzed progress in multimodal learning for radiology (Hartsock & Rasool, 2024; Li et al., 2023), they also expose risks of patient re-identification and data leakage through inference or inversion attacks (Shokri et al., 2017; Jagielski et al., 2020; Patil et al., 2025a). Prior defenses such as anonymization, differential privacy, and federated learning (McMahan et al., 2017) mitigate but do not eliminate these risks. Machine unlearning (Nguyen et al., 2025; Jang et al., 2023; Zhou et al., 2023; Liu et al., 2025a) offers a more targeted alternative that can act directly on trained models. However, existing unlearning methods and benchmarks assume flat data structures, overlooking the hierarchical organization of medical records. We address this gap with a hierarchy-aware unlearning benchmark and a method that explicitly models these structured dependencies.

## 3 Benchmark: MedForget

Here, we describe the hierarchical structure and unlearning design in Section 3.1. Section 3.2 describes the dataset construction, including data collection and task design. In Section 3.3, we explain the dataset partitioning strategy. Finally, we present key dataset statistics in Section 3.4.

### 3.1 Hierarchical Design and Unlearning Formulation

**Hierarchy Structure.** We define a clinically grounded hierarchy as: **Institution ⊃ Patient ⊃ Study ⊃ Section**, which is consistent with standard representations used in medical informatics (Johnson et al., 2019; Irvin et al., 2019; Banerjee et al., 2023), where clinical data are routinely organized along institution-patient-study-section hierarchies to capture diagnostic context and data provenance. Specifically, an *Institution* represents a healthcare provider or a site consisting of multiple patients. Each *Patient* includes one or more *Studies*, corresponding to distinct imaging visits or diagnostic episodes. Each *Study* comprises multimodal data (radiographs paired with textual reports), while each *Section* denotes a semantic component of the report (e.g., *Findings*, *Impression*). These entities form a nested structure where information flows upward: sections collectively describe a study, studies summarize a patient's trajectory, and patients are grouped under

institutions. This hierarchical organization induces structured dependencies among entities, implying that unlearning a node at any level must appropriately propagate to its descendants while preserving information associated with non-target siblings. These are challenges that flat data assumptions fail to capture.

**Unlearning Formulation.** Existing unlearning methods typically assume flat data with independent samples. However, clinical data exhibits hierarchical dependencies: forgetting a patient requires removing all associated studies and sections, while forgetting a single section should preserve sibling sections under the same study. We formalize this by defining unlearning at each hierarchy level: At the *Institution* level, unlearning should remove all patients, studies, and sections associated with target institutions. At the *Patient* level, unlearning should remove all studies and sections belonging to target patients. At the *Study* level, unlearning should remove all sections within target studies. Finally, at the *Section* level, unlearning should target only the specified sections, leaving all other hierarchical entities intact. At each level, non-target nodes should be preserved during unlearning. This formulation enables evaluation of both forgetting completeness and collateral damage. We detail concrete partition strategies in Section 3.3.

### 3.2 Data Construction Pipeline

**Data Collection.** We build MEDFORGET on top of MIMIC-CXR (Johnson et al., 2019), a large-scale de-identified multimodal medical dataset containing paired chest X-rays and radiology reports for over 65,000 patients. Each patient has one or more imaging studies, each study contains two chest X-ray views, and each report typically includes standardized sections such as *Examination*, *Indication*, *Findings*, and *Impression*, which form a natural *patient-study-section* hierarchy. We leverage this inherent structure to first retain only studies that contain at least three standard report sections, and then filter patients who have at least four such studies. To simulate a realistic multi-institutional environment, we group eligible patients into synthetic institutions with a *non-uniform* distribution: each institution is assigned between 4 and 12 patients, and each patient retains between 4 and 12 studies, reflecting natural variation in clinical practice. We randomly select eight such institutions to form MEDFORGET. We show just four institutions in Figure 3 for brevity.

**Task Design.** Building upon the hierarchical structure defined above, we design task types that evaluate how well models preserve or forget information across different semantic levels. While hierarchical unlearning determines *what* to forget, these tasks determine *how* forgetting impacts multimodal reasoning and comprehension. Specifically, for each report section, we define three complementary evaluation tasks, generation, cloze, and classification (see examples in Table 1), that capture distinct reasoning skills such as factual consistency, contextual inference, and cross-modal grounding. We synthetically generate task prompts and responses using the DeepSeek-V3 (DeepSeek-AI et al., 2025) model, grounded in authentic radiology text-image pairs, to enable efficient and semantically coherent medical text generation.

### 3.3 Dataset Partition

After generating the full set of multimodal question-answer pairs, we organize the data into structured subsets to support both fine-tuning and targeted unlearning experiments across different hierarchy levels. Below, we first describe the fine-tuning set used to obtain vanilla (pre-unlearning) MLLM performance, followed by the forget-retain partitions and evaluation sets. We provide examples from the dataset in Figure 2.

**Fine-tuning Set.** We construct question-answer pairs that explicitly encode both hierarchical context and section-level content. For the *Institution*, *Patient*, and *Study* levels, questions test entity identification (e.g., "From which institution does this medical image originate?") to enable cross-level understanding across the hierarchy. At the *Section* level, we include (prompt, response) pairs for all three task types. We then fine-tune the original instruction-tuned MLLM on this data to obtain the *vanilla MLLM*, whose parameters encode the hierarchical information.

**Forget-Retain Partitions.** To enable selective unlearning evaluation, we construct hierarchical forget-retain partitions (Figure 3) by designating 25% of entities as forget targets at each level of the hierarchy (as defined in Section 3.1) while the remaining entities under the same parent form the retain set. The Forget Set contains direct queries that the model should completely unlearn after the requested forgetting

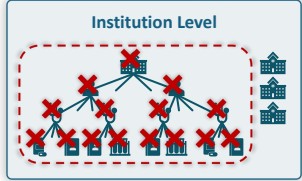 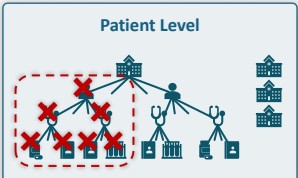 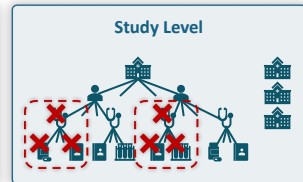 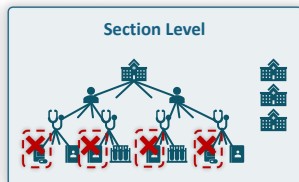

Figure 3: Illustration of the *forget-retain* partition at each hierarchy level, where approximately 25% of entities (red) form the forget set and the remainder constitute the retain set. This setup supports controlled multi-level unlearning experiments, capturing how forgetting propagates across hierarchically related data.

operation (e.g., "Looking at this medical image, what does the indication section indicate?" paired with sensitive content such as "woman status post resection of a large lung mass, with an interval chest X-ray (woman s/p rsxn of large lung mass // interval CXR)", see Figure 2 (a)). The Retain Set (Figure 2 (b)) consists of examples that must be preserved for the same forgetting request, typically medical knowledge from other patients, studies, or sections. A typical retain example asks the model to describe findings on an unrelated chest X-ray (e.g., "Resolution of right multifocal pneumonia. Unchanged left pleural effusion").

**Evaluation Sets.** To assess both forgetting performance and downstream clinical utility, apart from the forget and retain splits used during fine-tuning and unlearning, we evaluate models on two dedicated held-out sets. This structured evaluation protocol enables fine-grained analysis of whether models truly eliminate memorized information at the specified hierarchy level and retain their broader diagnostic reasoning abilities.

- **Forget rephrase set** uses the same underlying ground-truth answers as the Forget set but introduces paraphrased questions (generated using DeepSeek-V3) and visual augmentations (one of four SV-DRR (Xie et al., 2025) angular variations: $-30°$, $-15°$, $+15°$, $+30°$). These modifications test whether the model still recalls memorized information when queried with reworded prompts or slightly altered viewpoints, preventing trivial exact-match forgetting evaluation and exposing residual memorization.
- **General med set** is an independent medical VQA benchmark PMC-VQA (Zhang et al., 2023) used to measure the preservation of overall clinical utility. The questions and images are entirely unrelated to any institution, patient, study, or section involved in the forget or retain sets, for example, identifying fracture types on radiographs from unseen sources.

### 3.4 Dataset Statistics

MEDFORGET is organized across four hierarchy levels: 8 institutions with 4-12 patients each, 64 patients with 4-12 studies each, 512 studies (each containing two chest X-ray views, yielding 1,024 images), and 20,480 multimodal VQA pairs (20 per image). A 25% forget ratio is applied independently at each hierarchy level to reflect realistic deletion requests across different hierarchy levels, so the specific forget entities (and their associated images) differ across levels (see Figure 3). The pairs span three task categories: each image has 12 *generation*, 4 *cloze*, and 4 *classification* pairs. The 12 generation pairs combine 4 section-level questions covering report content with 8 hierarchical-generation questions identifying the study, patient, and institution. In terms of section composition, the dataset mirrors typical radiology reports: Impression (23.6%), Findings (22.9%), Technique (20.1%), Indication (15.7%), and Examination (13.5%) dominate, with History (2.9%) and other clinically rare types occurring infrequently (see Figure 4).

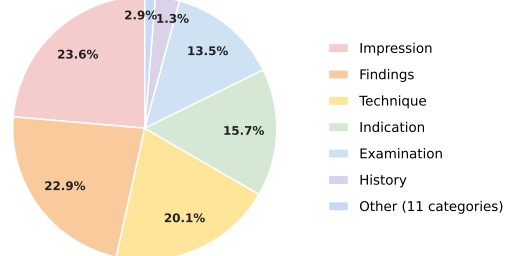

Figure 4: Distribution of section types in MEDFORGET.

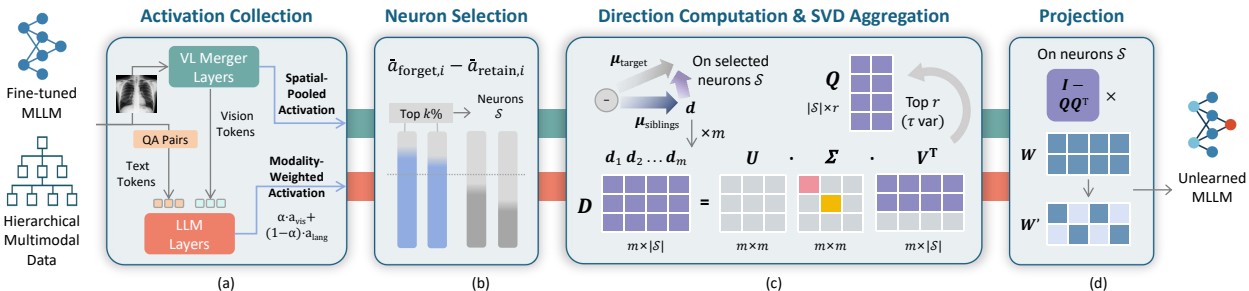

Figure 5: Illustration of CHIP. The method (a) collects cross-modal activations from both the vision-language merger and language layers, (b) identifies target-related neurons, (c) constructs hierarchy-aware directions that isolate sibling-differential information, and (d) applies a training-free subspace projection to remove these directions from both language layer weights and vision-language merger weights.

## 4    Method: CHIP

In this section, we discuss our proposed method CHIP (illustrated in Figure 5), a training-free approach for hierarchy-aware multimodal unlearning. Our method is motivated by a key observation: in hierarchical data, sibling nodes (i.e., nodes sharing a common parent) encode shared information reflecting their common context, while node-specific information is unique to each. Since the shared component is encoded across all siblings while the node-specific component is localized to each node, removing only the *sibling-differential* representations while preserving the *sibling-shared* component can enable targeted forgetting.

**Step 1: Cross-modal Activation Collection.** In multimodal models, node information is encoded across both the language backbone and vision projection layers[2]. Accordingly, for each forget or retain example, we feed the complete multimodal input, consisting of image tokens concatenated with the textual prompt and answer, into the model in a single forward pass. During this pass, we register forward hooks on the target layers, including the language feed-forward (FFN) layers and the vision–projection (merger) layers, to capture intermediate activations. We then aggregate these activations to produce one representative activation vector per sample per layer, which serves as the basis for subsequent neuron selection, direction computation, and model surgery.

For language layers, since the input sequence contains both vision tokens and text tokens, we separately compute the mean activation over vision-token positions and text-token positions, and combine them as:

$$\mathbf{a}^{(l)} = \alpha \cdot \mathbf{a}_{\text{vis}}^{(l)} + (1 - \alpha) \cdot \mathbf{a}_{\text{lang}}^{(l)}, \tag{1}$$

where $\mathbf{a}_{\text{vis}}^{(l)}$ and $\mathbf{a}_{\text{lang}}^{(l)}$ denote the mean activations over vision-token and text-token positions in layer $l$ (of hidden dimension $d_l$, one coordinate per neuron), respectively, and $\alpha$ controls their relative contribution. For merger layers, we apply global average pooling over the spatial (patch) dimension to obtain a single activation vector per sample. This design ensures that both modalities contribute appropriately to the computed directions (see Figure 5 (a)).

**Step 2: Neuron Selection.** Not all neurons contribute equally to storing forget-specific information. Inspired by Liu et al. (2025b), we identify the neurons most relevant to forgetting using neuron-level importance scores. Intuitively, a neuron is important for forgetting if it is substantially more activated on forget samples than on retain samples. Let $\bar{a}_{\text{forget},i}^{(l)}$ and $\bar{a}_{\text{retain},i}^{(l)}$ denote the mean absolute activation $|a_i^{(l)}|$ of neuron $i$ over forget and retain VQA samples, respectively. For each neuron $i$, we compute an importance score:

$$s_i^{(l)} = \bar{a}_{\text{forget},i}^{(l)} - \bar{a}_{\text{retain},i}^{(l)}, \tag{2}$$

and select the top $k\%$ neurons with the highest scores as the selected neuron set $\mathcal{S}^{(l)}$. These selected neurons define the subspace on which we apply the surgery (see Figure 5 (b)).

---

[2]We use the names "Vision projection layers" and "VL merger layers" interchangeably in this paper.

**Step 3: Direction Computation and SVD Aggregation.** We formalize the intuition of isolating target-specific information using *sibling-differential directions.* Let $\mathcal{G} = (\mathcal{V}, \mathcal{E})$ denote a hierarchical graph. For a target node $n_{\text{target}}$ with siblings sharing the same parent, let $\boldsymbol{\mu}_n^{(l)}$ be the mean activation at node $n$ in layer $l$ computed over the selected neurons $\mathcal{S}^{(l)}$, i.e., $\boldsymbol{\mu}_n^{(l)} \in \mathbb{R}^{|\mathcal{S}^{(l)}|}$. We compute the sibling-differential direction as:

$$\mathbf{d}^{(l)} = \text{normalize}\left(\boldsymbol{\mu}_{\text{target}}^{(l)} - \boldsymbol{\mu}_{\text{siblings}}^{(l)}\right), \tag{3}$$

where $\boldsymbol{\mu}_{\text{siblings}}^{(l)}$ denotes the mean activation over retained sibling nodes. By subtracting the sibling mean, the shared parent-level representation is factored out, isolating target-specific information in the direction.

For coarse-grained targets (e.g., institutions or patients), we decompose the target into all nodes within its hierarchical subtree and compute a separate sibling-differential direction for each node, contrasting it against the corresponding nodes drawn from the subtrees of the target's retained siblings. This *multi-direction decomposition* captures heterogeneous information across subtree nodes rather than collapsing it into a single averaged direction, yielding multiple directions per target (see Figure 5 (c)). We discuss the sensitivity of CHIP to the availability of sibling nodes in Appendix A.12.

We aggregate all directions associated with the subtree of a coarse-grained target using singular value decomposition (SVD) to obtain a low-rank subspace capturing dominant forget-specific variance. Let $\mathbf{D}^{(l)} \in \mathbb{R}^{m \times |\mathcal{S}^{(l)}|}$ be the matrix of $m$ stacked directions for layer $l$. We compute $\mathbf{D}^{(l)} = \mathbf{U}\boldsymbol{\Sigma}\mathbf{V}^\top$ and retain the top $r$ right singular vectors corresponding to the largest singular values that cumulatively explain at least $\tau$ (e.g., 95%) of the total variance, forming the projection basis $\mathbf{Q}^{(l)} \in \mathbb{R}^{|\mathcal{S}^{(l)}| \times r}$.

**Step 4: Weight Subspace Multimodal Projection.** We then update the weights corresponding to the selected neurons via orthogonal projection. Motivated by Belrose et al. (2023); Kodge et al. (2024), we perform the weight update as:

$$\mathbf{W}_{\mathcal{S},:}^{(l)} \leftarrow \left(\mathbf{I} - \mathbf{Q}^{(l)}(\mathbf{Q}^{(l)})^\top\right) \mathbf{W}_{\mathcal{S},:}^{(l)}, \tag{4}$$

where $\mathbf{W}_{\mathcal{S},:}^{(l)}$ denotes the rows of the weight matrix corresponding to the selected neurons, $\mathbf{W}_{\mathcal{S},:}^{(l)} \in \mathbb{R}^{|\mathcal{S}^{(l)}| \times d_{\text{in}}}$. This projection removes components aligned with the forget-specific subspace while preserving orthogonal information (see Section A.4 for detailed derivations). Applying surgery only to language layers risks leaving forget-specific information in the merger weights, which continue to project relevant visual features into the language space. CHIP therefore applies surgery jointly to upper language layers and the vision-merger layers in the MLLM, removing forget-specific representations at both stages (see Figure 5 (d)). We choose layers for this weight projection empirically and provide the ablation results in Appendix A.10.

# 5 Experiments

## 5.1 Experimental Setup

**Evaluation Metrics.** We evaluate unlearning performance using three complementary metrics that capture both forgetting completeness and retain set utility across the three tasks: (1) **Generation Score (Gen Score)** for generation tasks, computed as a weighted average of 75% factuality score and 25% ROUGE-L (Lin, 2004), reflecting the prioritization of semantic correctness over surface-level lexical overlap in clinical text evaluation. The factuality score is obtained by using GPT-4o (Hurst et al., 2024) as LLM-as-a-judge (Gu et al., 2026) to rate clinical accuracy on a 1–10 scale, following prior work (Sun et al., 2023; Yu et al., 2024; Zheng et al., 2023); To verify that our conclusions are not sensitive to the choice of LLM judge, we additionally report the full evaluation using DeepSeek-V3 as the factuality judge in Table 9 and Appendix A.2, where the two judges show strong agreement (Pearson $r = 0.9993$, Spearman $\rho = 0.9987$). (2) **Cloze Accuracy (Cloze Acc)**, measured by exact string match in cloze-style completion tasks; and (3) **Classification Accuracy (Class. Acc)**, computed as the proportion of correct multiple-choice predictions given the question and chest X-ray image. Evaluation metric details are provided in Appendix A.5. We also provide a detailed discussion of the F/R Diff metric in Appendix A.5.

| Hierarchy | Method | Forget Set ↓ | | | Retain Set ↑ | | | F/R Diff ↑ | Evaluation Sets | | | | | |
|---|---|---|---|---|---|---|---|---|---|---|---|---|---|---|
| | | | | | | | | | Forget Rephrase Set ↓ | | | General Med Set ↑ | | |
| | | Gen Score | Class. Acc | Cloze Acc | Gen Score | Class. Acc | Cloze Acc | | Gen Score | Class. Acc | Cloze Acc | Gen Score | Class. Acc | Cloze Acc |
| Section | Vanilla | 98.7 | 99.6 | 99.2 | 99.3 | 100.0 | 98.5 | 0.6 | 54.6 | 99.7 | 92.1 | 67.3 | 95.1 | 83.7 |
| | MANU | 55.1 | 99.8 | 95.0 | 56.1 | 98.3 | 90.4 | 1.0 | 50.6 | 99.5 | 89.7 | 57.1 | 85.3 | 74.1 |
| | Grad. Diff. | 55.3 | 99.9 | 97.2 | 58.8 | 98.7 | 88.9 | 3.5 | 50.3 | 99.7 | 85.8 | 62.0 | 84.6 | 75.7 |
| | KL Min. | 54.0 | 99.7 | 96.7 | 56.2 | 99.3 | 92.5 | 2.2 | 52.6 | 99.6 | 89.1 | 61.4 | 84.3 | 73.6 |
| | NPO | 55.7 | 99.6 | 97.1 | 57.1 | 99.7 | 91.4 | 1.4 | 54.2 | 99.4 | 89.3 | 62.9 | 85.6 | 75.9 |
| | **CHIP** | **51.8** | **99.4** | **93.5** | 60.1 | 100.0 | 93.9 | 8.3 | **46.9** | **99.2** | **84.9** | 64.2 | 88.2 | 78.6 |
| Study | Vanilla | 97.6 | 99.8 | 99.5 | 98.1 | 99.8 | 97.8 | 0.5 | 51.6 | 99.8 | 95.1 | 68.4 | 95.8 | 84.3 |
| | MANU | 41.0 | 99.9 | 90.5 | 45.4 | 99.1 | 84.5 | 4.4 | 37.9 | 99.3 | 80.6 | 56.4 | 87.3 | 75.9 |
| | Grad. Diff. | 41.8 | 99.5 | 86.8 | 50.4 | 99.0 | 82.1 | 8.6 | 39.1 | 99.6 | 79.2 | 61.2 | 84.4 | 73.4 |
| | KL Min. | 45.5 | 99.7 | 89.0 | 47.4 | 99.8 | 89.0 | 1.9 | 40.1 | 99.4 | 79.8 | 60.8 | 86.4 | 76.8 |
| | NPO | 44.6 | 99.8 | 90.2 | 46.2 | 99.6 | 85.7 | 1.6 | 41.7 | 99.5 | 79.3 | 62.1 | 83.8 | 74.9 |
| | **CHIP** | **39.6** | **99.1** | **86.2** | 49.8 | 100.0 | 88.1 | 10.2 | **35.0** | **99.2** | **77.0** | 63.6 | 88.6 | 78.0 |
| Patient | Vanilla | 99.2 | 100.0 | 98.2 | 99.4 | 99.9 | 99.6 | 0.2 | 64.3 | 99.9 | 92.8 | 57.7 | 94.4 | 84.0 |
| | MANU | 42.9 | 97.1 | 83.8 | 52.2 | 98.4 | 78.5 | 9.3 | 36.2 | 99.7 | 79.4 | 31.6 | 83.3 | 67.4 |
| | Grad. Diff. | 42.7 | 96.7 | 82.5 | 52.3 | 97.6 | 75.7 | 9.6 | 36.9 | 99.8 | 77.2 | 37.9 | 88.4 | 74.9 |
| | KL Min. | 47.3 | 97.4 | 81.0 | 48.3 | 99.8 | 82.2 | 1.0 | 39.3 | 99.5 | 79.1 | 38.3 | 88.8 | 73.7 |
| | NPO | 45.3 | 96.9 | 81.6 | 49.8 | 98.8 | 75.5 | 4.5 | 41.1 | 99.6 | 78.2 | 35.4 | 83.4 | 71.8 |
| | **CHIP** | **41.7** | **96.4** | **79.9** | 53.3 | 99.3 | 81.2 | 11.6 | **33.7** | **99.3** | **75.0** | 36.3 | 88.1 | 73.0 |
| Institution | Vanilla | 98.1 | 99.6 | 98.7 | 98.3 | 99.8 | 99.8 | 0.2 | 59.6 | 100.0 | 96.5 | 60.3 | 93.1 | 83.8 |
| | Retrained | 19.1 | 68.6 | 53.4 | 97.6 | 99.7 | 99.5 | 78.5 | 19.8 | 73.1 | 54.6 | 59.4 | 92.4 | 83.2 |
| | MANU | 43.0 | 99.1 | 81.4 | 48.8 | 97.6 | 78.3 | 5.8 | 34.2 | 99.8 | 74.4 | 52.1 | 84.0 | 71.1 |
| | Grad. Diff. | 42.2 | 98.8 | 80.1 | 53.8 | 98.3 | 77.3 | 11.6 | 36.4 | 99.6 | 70.3 | 59.7 | 88.5 | 77.1 |
| | KL Min. | 43.4 | 98.5 | 79.9 | 47.7 | 99.8 | 78.9 | 4.3 | 39.5 | 99.5 | 73.8 | 58.7 | 88.0 | 76.4 |
| | NPO | 44.7 | 98.9 | 80.3 | 47.5 | 98.0 | 77.6 | 2.8 | 38.5 | 99.4 | 74.9 | 56.2 | 82.6 | 73.2 |
| | **CHIP** | **38.6** | **98.2** | **76.9** | 53.2 | 100.0 | 80.2 | 14.6 | **32.0** | **99.2** | **69.7** | 57.5 | 87.4 | 75.2 |

Table 2: Performance across hierarchical unlearning splits and tasks. All values are percentages (%). F/R Diff denotes the difference between Retain Set and Forget Set Gen Scores, measuring the unlearning-utility gap. Retrained denotes the gold-standard model retrained from scratch only on the retain set, reported at the Institution level. ↓ indicates lower is better for unlearning (forget and its rephrase sets), ↑ indicates higher is better for utility preservation (retain, F/R Diff, and general sets). Best unlearning and utility performance are shown in **bold** and underlined. Results are averaged over three random evaluation seeds; standard deviations are omitted as they are consistently below 0.2%.

**Compared Unlearning Methods.** We benchmark four representative unlearning methods on MED-FORGET: (1) Gradient Difference (Grad. Diff.) (Yao et al., 2024), (2) Negative Preference Optimization (NPO) (Zhang et al., 2024), (3) Modality-Aware Neuron Unlearning (MANU) (Liu et al., 2025b), and (4) KL Minimization (KL Min.) (Nguyen et al., 2020). We provide more details in Appendix A.3.

**Unlearning Pipeline.** We evaluate unlearning methods using a multi-stage pipeline on MEDFORGET. First, we fine-tune Lingshu-7B (Xu et al., 2025), a Qwen2.5-VL-based medical MLLM using LoRA (Hu et al., 2022), on the full hierarchical training set to obtain the vanilla model. We also provide results on MedGemma-4B in Appendix A.1. For each hierarchy level, we construct corresponding forget and retain splits (Section 3.3), and apply each unlearning method independently to the vanilla checkpoint, producing four unlearned variants per method (one per hierarchy level). Forgetting at higher hierarchy levels subsumes all subordinate nodes. We provide more details in Appendix A.6.

**Implementation Details.** We fine-tune Lingshu-7B using LoRA with rank $r = 16$ and $\alpha = 32$, trained with AdamW optimizer (learning rate $1 \times 10^{-4}$, batch size 8, 3 epochs). For training-based unlearning methods (Grad. Diff., NPO, KL Min.), we use a learning rate $2 \times 10^{-5}$, batch size 4, and 1 epoch. The training-free methods, MANU and CHIP, require no gradient updates; CHIP uses $\alpha = 0.3$ and $\tau = 0.95$ across all hierarchy levels. We run baseline methods on two NVIDIA L40S GPUs (48 GB each), while CHIP requires only a single L40S GPU. We provide method-specific hyperparameters in Appendix A.7 for reproducibility and discuss sensitivity to hyperparameters in Appendix A.13 and the choice of fine-tuning strategy in Appendix A.11.

## 5.2 Main Results

### 5.2.1 Comparison of Deletion-Utility Trade-off with Baselines

As shown in Table 2, existing unlearning methods struggle to balance effective forgetting with utility preservation. Training-based methods (Grad. Diff., KL Min., and NPO) maintain relatively strong retain perfor-

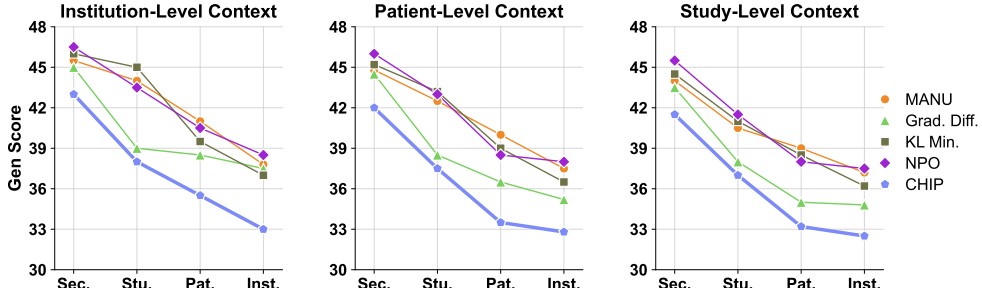

Figure 6: Section-level leakage under hierarchical reconstruction attacks. Each subplot corresponds to a different level of contextual prompting (Institution, Patient, or Study), where an adversary progressively includes higher-level identifiers to recover forgotten Section-level content. The x-axis denotes the granularity at which unlearning was performed, where Inst., Pat., Stu., and Sec. refer to Institution, Patient, Study, and Section, respectively. Lower Gen Scores indicate stronger resistance to leakage.

mance (e.g., retain Gen Scores of 47.5–53.8 at the Institution level) but leave substantial residual memorization on the forget set (forget Gen Scores $\geq$ 42.2). MANU, despite being training-free, does not achieve lower forget scores (43.0) than training-based baselines and incurs noticeably larger drops in retain (48.8) and general medical performance (52.1). In contrast, CHIP consistently achieves a stronger deletion-utility trade-off across hierarchies. For example, at the Institution level, it attains the lowest forget Gen Score (38.6) while maintaining a high retain Gen Score (53.2), yielding the largest F/R Diff (14.6, compared to 11.6 for Grad. Diff.). Similar trends hold at other levels, where CHIP achieves lower forget scores while preserving competitive retain performance under hierarchy-aware multimodal unlearning. Additionally, CHIP demonstrates improved robustness on the forget rephrase set (e.g., 32.0 at Institution level vs. 36.4 for Grad. Diff.) while remaining competitive on the general medical set. These results highlight CHIP's effectiveness in achieving a superior deletion-utility balance under hierarchy-aware multimodal unlearning.

### 5.2.2 Performance Across Tasks

Table 2 shows that achieving consistent forgetting across heterogeneous tasks is challenging. Cloze and classification tasks are more tightly coupled to core representations than free-form generation, making them harder to unlearn without degrading retained generation quality. We therefore tune all methods so that retain-set generation scores fall within a comparable range (approximately 45–60), and select hyperparameters that maximize the forget-retain performance gap under this constraint. This ensures a fair comparison by approximately matching utility preservation across methods. At comparable retain generation levels, CHIP consistently achieves lower forget-set cloze and generation scores across hierarchies, with modest reductions in classification accuracy, while maintaining competitive retain and general medical performance. This demonstrates CHIP's more task-robust unlearning, with improved suppression of task-specific memorization beyond surface-level degradation. All unlearning methods including CHIP, still exhibit high forget-set classification accuracy, indicating incomplete removal of discriminative signals. We discuss the cause of high forget-set classification accuracy after unlearning in Appendix A.14.

### 5.2.3 Resistance to Hierarchical Reconstruction Attacks

We evaluate whether unlearned models remain vulnerable to reconstruction when an adversary exploits hierarchical context. We simulate a cumulative reconstruction attack that progressively adds identifiers (Institution, Patient, Study) to the prompt while attempting to regenerate forgotten Section-level content (see Table 6 in the Appendix for examples of the attack prompt). Leakage is measured using the Gen Score, where higher values indicate greater recovery risk. As shown in Figure 6, models unlearned only at fine granularity (Section or Study) remain highly vulnerable, often regenerating forgotten content with high fidelity (Gen Score between 36–47) under strong contextual prompts. In contrast, higher-granularity unlearning (Patient or Institution) substantially improves robustness, reducing Gen Score to 33–41. Across

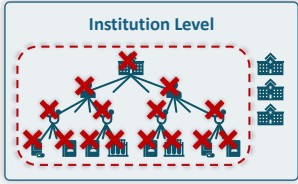 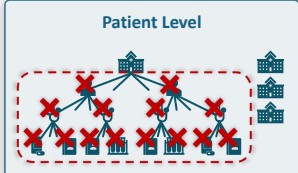 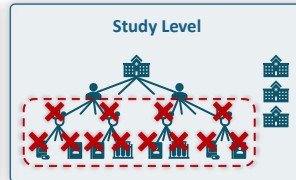 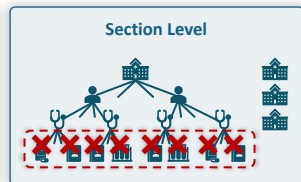

Figure 7: Illustration of the hierarchy-consistent *forget-retain* split used in the granularity ablation. Unlike the standard partition (Figure 3), where forget targets are sampled independently at each level, here the section-level split is held constant across all four hierarchy levels. The four settings differ only in the hierarchical context during forgetting, ranging from full context (Institution level) to none (Section level).

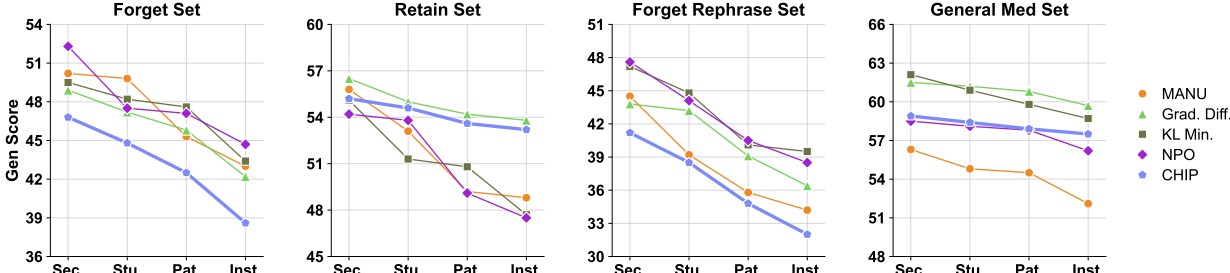

Figure 8: Impact of forgetting granularity with a fixed section-level forget-retain split. Performance of unlearning methods on the Forget, Retain, Forget Rephrase, and General Med sets as the forgetting granularity varies from Section to Institution level. Unlike the standard setting (Figure 3), the section-level split is held constant across all levels, isolating the effect of hierarchical context on the forgetting-utility trade-off. The x-axis labels denote: Inst. (Institution), Pat. (Patient), Stu. (Study), and Sec. (Section).

all methods and hierarchical levels, CHIP achieves the lowest leakage (see Figure 6) among all compared methods, reaching as low as 32.5 at Institution-level unlearning, demonstrating its robustness to hierarchical reconstruction attacks.

### 5.2.4 Impact of Hierarchical Granularity

To isolate the effect of forgetting granularity, we design a controlled experiment in which the section-level forget-retain split is held constant across all hierarchy levels. Specifically, we select 25% of institutions along with a subset of their patients, associated studies, and sections as the shared forget targets. We then perform forgetting at each of the four hierarchy levels over the same split, with the only difference being the hierarchical context included in the forgetting operation. As illustrated in Figure 7, Institution-level forgetting now incorporates institution, patient, and study identifiers; Patient-level forgetting includes patient and study identifiers; Study-level forgetting uses only study identifiers; and Section-level forgetting targets sections directly without any higher-level context. This design ensures that the sole variable across the levels is the granularity of the forgetting request, enabling a direct comparison of how hierarchical granularity affects the forgetting-utility trade-off.

Our results in Figure 8 show that unlearning effectiveness varies systematically with hierarchical granularity. Coarse-grained deletion enables stronger forgetting but induces larger utility loss: at the Institution level, forget Gen Scores drop to 42.2–44.7 across baseline methods, while retain Gen Scores decline to 47.5–53.8. At finer granularity, forgetting becomes harder, but utility is better preserved: at the Section level, retain Gen Scores remain high (54.2–56.5), while forget Gen Scores rise to 48.9–52.3, indicating residual memorization. Across all hierarchies, CHIP achieves favorable trade-offs, attaining the lowest forget Gen Scores at coarse levels (e.g., 38.6 at the Institution level) while maintaining competitive retain performance (53.2). It also preserves utility at the Section level (55.2) with only modest increases in forget scores (46.8). The results highlight a hierarchy-dependent deletion-utility trade-off and show CHIP's robustness across granularities.

# 6 Conclusion

This work exposes the limitations of flat unlearning in multimodal medical models and motivates the need for hierarchy-aware unlearning. We introduce MEDFORGET, the first hierarchical benchmark for medical unlearning, which reveals privacy–utility trade-offs unaddressed by existing methods. We further propose CHIP, a training-free approach that selectively removes target-specific information while preserving shared representations, achieving stronger forgetting across hierarchies and improved robustness to reconstruction attacks. Together, our benchmark and method advance the deployment of trustworthy medical MLLMs.

**Broader Impact Statement**

MEDFORGET addresses critical privacy challenges in medical AI by introducing the first comprehensive benchmark for hierarchy-aware unlearning in multimodal clinical models. By explicitly modeling realistic medical hierarchies (e.g., institution, patient, study, and section), our benchmark supports the development and evaluation of unlearning methods aligned with regulatory requirements such as HIPAA (Annas, 2003) and GDPR (Voigt & Von dem Bussche, 2017), including the "right to be forgotten." We therefore view this work as a step toward safer medical AI systems, complementing, rather than replacing, other safeguards such as access control, auditability, and data minimization. Importantly, all evaluated methods still leave substantial residual memorization, and thus should not be interpreted as providing regulatory-grade erasure.

**Acknowledgments**

We would like to thank Elias Stengel-Eskin for his feedback on a draft of the paper. This work was supported by National Institutes of Health (NIH) under other transactions 1OT2OD038045-01, ARO Award W911NF2110220, and ONR Grant N00014-23-1-2356. The views contained in this article are those of the authors and not of the funding agency.

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

## A Appendix

### A.1 Generalization Results on MedGemma-4B

To evaluate the generalizability of our findings beyond Lingshu-7B, we repeat the full unlearning evaluation pipeline using MedGemma-4B (Sellergren et al., 2025), a 4B-parameter Gemma-3-based medical MLLM for medical text and image comprehension. We fine-tune MedGemma-4B under the same LoRA configuration and apply all five unlearning methods independently at each hierarchy level. Results are reported in Table 3 and Table 10 (extended metrics). The key findings from the Lingshu-7B (Qwen-2.5-VL-based model) evaluation are consistently reproduced on MedGemma. CHIP achieves the highest F/R Diff across all four hierarchy levels (6.7 at Section, 7.0 at Study, 14.4 at Patient, and 11.9 at Institution), while baseline methods consistently yield lower F/R Diff than CHIP across all levels. The cross-task patterns also hold: classification accuracy on the forget set remains high for all methods, and CHIP achieves the lowest forget-set cloze and classification accuracy at most levels. This cross-architecture consistency confirms that both the challenges exposed by MEDFORGET and the effectiveness of CHIP generalize beyond a specific model family.

### A.2 Evaluation with DeepSeek-V3

To verify that our conclusions are not sensitive to the choice of LLM judge, we additionally rerun the full evaluation using DeepSeek-V3 as the factuality judge. We report the results are in Table 9. We observe the same qualitative trends across all hierarchy levels and methods, including the relative ranking of baselines and the superior deletion-utility trade-off achieved by CHIP. Quantitatively, the scores produced by GPT-4o and DeepSeek-V3 judges are highly correlated, with Pearson correlation $r = 0.9993$ and Spearman correlation $\rho = 0.9987$. These results suggest that our findings are robust to the choice of evaluator and are not driven by a particular LLM judge.

### A.3 Compared Unlearning Methods

We benchmark multiple unlearning strategies spanning prompt-based and gradient-based paradigms. Specifically, we evaluate:

**Gradient Difference (Grad. Diff.)** (Yao et al., 2024): It is a version of gradient ascent that optimizes the forget and retain objectives. Grad. Diff. introduces a balanced objective that simultaneously decreases loss on the retain set. The objective is given by

$$\mathcal{L}_{\text{diff}} = -\mathcal{L}(S_F, w) + \mathcal{L}(S_R, w),$$

where $S_F$ and $S_R$ denote the forget and retain sets, respectively. We interleave samples from both sets for computational efficiency while ensuring that deletion does not damage model utility on the retain data.

**KL Minimization (KL Min.)** (Nguyen et al., 2020): It is an unlearning technique that is guided by divergence and aims for targeted forgetting on the forget set $S_F$. It simultaneously aims to anchor the model's outputs on the retain set $S_R$ to those of the original fine-tuned model. Its approach aims to maximize the cross-entropy loss on $S_F$ for deletion and aims to minimize the KL divergence on $S_R$ to get high retain set performance. The composite objective is given by

$$\mathcal{L}_{\text{KL}} = -\mathcal{L}(S_F, w)$$
$$+ \frac{1}{|S_R|} \sum_{s \in S_R} \text{KL}(p_{\text{o}}(s) \parallel p_{\text{c}}(s)),$$

where $p_{\text{o}}$ and $p_{\text{c}}$ denote the output distributions of the original and current models, respectively, and $w$ represents the model parameters. By interleaving batches from both sets during optimization, this method achieves efficient unlearning with preserved utility on unrelated data.

**Negative Preference Optimization (NPO)** (Zhang et al., 2024): It is a preference optimization method designed to reduce the instability in gradient ascent. NPO treats forget set samples as negative preferences,

| Hierarchy | Method | Forget Set ↓ | | | Retain Set ↑ | | | F/R Diff ↑ | Evaluation Sets | | | | | |
| | | | | | | | | | Forget Rephrase Set ↓ | | | General Med Set ↑ | | |
| | | Gen Score | Class. Acc | Cloze Acc | Gen Score | Class. Acc | Cloze Acc | | Gen Score | Class. Acc | Cloze Acc | Gen Score | Class. Acc | Cloze Acc |
|---|---|---|---|---|---|---|---|---|---|---|---|---|---|---|
| Section | Vanilla | 98.3 | 100.0 | 98.8 | 98.8 | 99.8 | 99.3 | 0.5 | 60.6 | 99.7 | 92.3 | 69.0 | 93.9 | 82.7 |
| | MANU | 55.4 | 99.8 | 94.1 | 56.3 | 99.5 | 88.4 | 0.9 | 50.4 | 99.6 | 85.6 | 58.8 | 84.9 | 76.9 |
| | Grad. Diff. | 55.9 | 99.9 | 95.0 | 57.1 | 99.1 | 93.0 | 1.2 | 50.1 | 99.8 | 87.3 | 63.2 | 85.9 | 73.4 |
| | KL Min. | 56.5 | 99.6 | 95.6 | 57.0 | 99.9 | 95.8 | 0.5 | 53.5 | 99.5 | 87.4 | 64.3 | 83.8 | 73.9 |
| | NPO | 57.2 | 99.7 | 96.0 | 57.9 | 98.0 | 92.1 | 0.7 | 53.6 | 99.9 | 86.2 | 63.1 | 83.5 | 73.8 |
| | **CHIP** | **53.3** | **99.3** | **92.8** | 60.0 | 100.0 | 94.0 | 6.7 | **47.7** | **99.3** | **84.0** | 65.7 | 88.0 | 78.0 |
| Study | Vanilla | 97.5 | 100.0 | 97.8 | 98.7 | 99.7 | 99.3 | 1.2 | 52.3 | 99.6 | 92.1 | 70.5 | 92.9 | 87.8 |
| | MANU | 39.8 | 99.6 | 87.4 | 40.8 | 98.6 | 84.4 | 1.0 | 34.4 | 99.4 | 79.8 | 56.0 | 84.6 | 75.6 |
| | Grad. Diff. | 37.0 | 100.0 | 87.1 | 41.7 | 97.8 | 82.3 | 4.7 | 36.0 | 99.7 | 78.5 | 60.9 | 85.9 | 73.0 |
| | KL Min. | 39.3 | 99.7 | 87.2 | 40.5 | 97.4 | 85.5 | 1.2 | 38.1 | 99.8 | 79.0 | 60.7 | 83.8 | 75.5 |
| | NPO | 40.2 | 99.9 | 87.5 | 41.5 | 97.5 | 81.6 | 1.3 | 39.6 | 99.5 | 78.2 | 61.5 | 82.9 | 75.1 |
| | **CHIP** | **36.4** | **98.8** | **85.5** | 43.4 | 99.8 | 87.5 | 7.0 | **31.8** | **99.3** | **76.0** | 63.6 | 87.5 | 77.5 |
| Patient | Vanilla | 99.0 | 99.6 | 97.5 | 99.5 | 99.9 | 98.1 | 0.5 | 67.6 | 99.9 | 94.4 | 71.4 | 94.4 | 83.4 |
| | MANU | 46.1 | 96.6 | 80.2 | 51.2 | 97.6 | 76.7 | 5.1 | 37.0 | 99.8 | 76.1 | 29.9 | 84.8 | 71.0 |
| | Grad. Diff. | 43.9 | 96.4 | 80.1 | 55.9 | 98.7 | 78.3 | 12.0 | 36.5 | 99.4 | 75.8 | 37.2 | 88.3 | 74.4 |
| | KL Min. | 44.1 | 96.7 | 82.8 | 51.3 | 98.2 | 79.6 | 7.2 | 38.9 | 99.6 | 77.5 | 35.5 | 89.3 | 74.3 |
| | NPO | 44.4 | 96.9 | 83.2 | 50.3 | 97.2 | 77.4 | 5.9 | 41.7 | 99.7 | 78.8 | 33.7 | 84.1 | 68.2 |
| | **CHIP** | **40.6** | **96.1** | **78.6** | 55.0 | 99.5 | 81.0 | 14.4 | **33.8** | **99.2** | **74.0** | 34.8 | 87.8 | 72.5 |
| Institution | Vanilla | 98.2 | 99.5 | 98.1 | 98.4 | 99.9 | 98.4 | 0.2 | 55.1 | 100.0 | 96.2 | 61.1 | 93.7 | 86.7 |
| | MANU | 44.5 | 98.6 | 78.6 | 49.2 | 99.4 | 78.5 | 4.7 | 41.8 | 99.9 | 70.3 | 53.5 | 82.1 | 70.9 |
| | Grad. Diff. | 44.9 | 98.7 | 77.8 | 55.0 | 97.5 | 74.8 | 10.1 | 42.1 | 99.7 | 71.0 | 60.6 | 87.8 | 75.8 |
| | KL Min. | 47.4 | 98.4 | 77.2 | 49.7 | 100.0 | 80.9 | 2.3 | 44.3 | 99.5 | 69.3 | 59.4 | 87.6 | 75.2 |
| | NPO | 45.8 | 98.9 | 77.0 | 49.6 | 98.8 | 77.9 | 3.8 | 44.6 | 99.4 | 70.9 | 56.9 | 83.6 | 70.0 |
| | **CHIP** | **41.5** | **97.8** | **75.8** | 53.4 | 99.8 | 80.0 | 11.9 | **38.3** | **99.1** | **68.0** | 58.9 | 86.8 | 74.5 |

Table 3: Performance across hierarchical unlearning splits and tasks for the MedGemma-4B model. All values are percentages (%). F/R Diff denotes the difference between Retain Set and Forget Set Gen Scores, measuring the unlearning-utility gap. ↓ indicates lower is better for unlearning (forget and its rephrase sets), ↑ indicates higher is better for utility preservation (retain, F/R Diff, and general sets). Best unlearning and utility performance are shown in **bold** and underlined.

encouraging the model to downweigh their influence relative to a reference policy derived from the retain set. The loss is

$$\mathcal{L}_{\text{NPO}} = \frac{2}{\beta} \mathbb{E}_{(x,y) \sim S_F} \big[ \log \big( 1 + r(x,y)^{\beta} \big) \big],$$

where $r(x,y) = \pi_\theta(y|x)/\pi_{\text{ref}}(y|x)$, with $\beta$ controlling the optimization curvature and $\pi_{\text{ref}}$ as the retain-only reference model. This yields smoother parameter updates, averting the performance collapse typical of unregularized ascent methods.

**Modality-Aware Neuron Unlearning (MANU)** (Liu et al., 2025b): It is a neuron-level pruning framework for MLLM unlearning that selectively removes modality-specific knowledge contributions. It works in two stages: (1) *important neuron selection*, where it computes an importance score $\mathcal{I}(\mathcal{D}, n)$ for each neuron $n$ across both modalities using four metrics, absolute ($I_{\text{abs}}$), frequency ($I_{\text{freq}}$), variance ($I_{\text{var}}$), and root mean square ($I_{\text{rms}}$), defined on activation deviations between textual and multimodal subsets of $\mathcal{D}$; and (2) *selective pruning*, which ranks neurons by the ratio score $S_n = \mathcal{I}(\mathcal{D}_f, n)/(\mathcal{I}(\mathcal{D}_r, n) + \epsilon)$ and sets weights to zero for the top $\alpha\%$ most forget-associated neurons. By disentangling modality-specific activations, it aims to enable targeted forgetting across both modalities, achieving balanced unlearning.

## A.4 Orthogonal Projection Derivation

**Mathematical Formulation.** Given the projection basis $\mathbf{Q}^{(l)} \in \mathbb{R}^{|\mathcal{S}^{(l)}| \times r}$ obtained from SVD, where columns of $\mathbf{Q}^{(l)}$ are orthonormal (i.e., $(\mathbf{Q}^{(l)})^\top \mathbf{Q}^{(l)} = \mathbf{I}_r$), the matrix $\mathbf{P} = \mathbf{Q}^{(l)}(\mathbf{Q}^{(l)})^\top$ is the orthogonal projection matrix onto the subspace spanned by the columns of $\mathbf{Q}^{(l)}$.

The complementary projection $\mathbf{P}^\perp = \mathbf{I} - \mathbf{Q}^{(l)}(\mathbf{Q}^{(l)})^\top$ projects onto the orthogonal complement of this subspace. Applying $\mathbf{P}^\perp$ to the weight matrix rows:

$$\mathbf{W}_{\mathcal{S},:}^{(l)} \leftarrow \Big( \mathbf{I} - \mathbf{Q}^{(l)}(\mathbf{Q}^{(l)})^\top \Big) \mathbf{W}_{\mathcal{S},:}^{(l)} \tag{5}$$

removes all components of the weight vectors that lie within the forget-specific subspace, while preserving components orthogonal to it.

| Dataset | Data Structure | Context Type | Task Types | Unlearning Relevance |
|---|---|---|---|---|
| **MLLMU-Bench** (Liu et al., 2025a) | Single image; single long context | Person profile or caption | Generation, classification, and cloze-style tasks | High — tests multimodal memorization and recall, but lacks hierarchical or compositional structure. |
| **CLEAR** (Dontsov et al., 2025) | Multiple images; multiple short contexts | Image captions | Name recognition; entity prediction | Medium — captures entity-level leakage, but no hierarchical or multimodal reasoning. |
| **UnLOK-VQA** (Patil et al., 2024b) | Single image; single question | Pretrained knowledge only | VQA-style entity prediction | Low — focuses on object-level forgetting, without modeling dataset-scale deletions. |
| MEDFORGET **(Ours)** | **Hierarchical:** Institution → Patient → Study → Section | **Multimodal:** Clinical images + text reports | **Generation, classification, and cloze-style tasks** across hierarchy levels | **High** — supports hierarchical unlearning that mirrors real-world medical deletion requests across multiple granularity levels. |

Table 4: Comparison of multimodal unlearning benchmarks. Prior multimodal unlearning datasets study shallow or flat structures with limited contextual dependencies. MEDFORGET introduces clinically grounded, hierarchically structured data that spans institutions, patients, and studies. It combines multimodal (image-text) reasoning with hierarchical unlearning challenges, enabling realistic assessment of unlearning performance at different granularities.

**Connection to Prior Work.** This orthogonal projection technique for removing specific information from neural network parameters has been explored in prior work. Belrose et al. (2023) introduced LEACE for closed-form concept erasure in representation space, proving that projecting onto the nullspace of concept-related directions prevents linear classifiers from recovering the erased concept. Kodge et al. (2024) applied a similar principle directly to weight matrices for class unlearning, using SVD to identify class-discriminatory subspaces and projecting weights onto their orthogonal complement.

Our method builds upon this foundation but differs in the construction of the projection basis: rather than using simple forget-vs-retain activation differences, we compute *sibling-differential directions* that exploit hierarchical graph structure to isolate target-specific representations while canceling shared semantic information. Additionally, we apply selective neuron surgery and jointly operate across language and vision-language merger layers.

### A.5 Evaluation Metrics

We evaluate all unlearning methods using three complementary metrics, each tailored to assess forgetting completeness and utility preservation in medical MLLMs:

**Generation Score (Gen Score)** for generation tasks, computed as a weighted average of 75% factuality score and 25% ROUGE-L score (Lin, 2004). The *factuality score* is obtained by prompting GPT-4o (Hurst et al., 2024) to rate the factual accuracy and medical consistency of each generated clinical text against the ground truth on a 1–10 scale (1 = nonsensical or clinically incorrect; 10 = fully accurate and consistent), following approaches in (Sun et al., 2023; Yu et al., 2024; Zheng et al., 2023). The complete evaluation prompt template is provided in Table 5. The *ROUGE-L score* captures the longest-common-subsequence overlap between generated and reference texts, reflecting both precision and recall in clinical narrative generation.

---

**Factuality Score Evaluation Prompt**

You will evaluate the factuality of the "generated_answer" against the "ground_truth" for medical image analysis questions. Assess how well the response captures the KEY MEDICAL INFORMATION and assign a score (1–10).

**Evaluation Principles:**

1. Medical Terminology Accuracy

2. Core Clinical Content (anatomical structures, findings, diagnostic information)

3. Partial Credit for key concepts

4. Context Sensitivity

**Scoring Rubric:**
**10–9:** Fully correct; all key findings present
**8–7:** Mostly correct; minor omissions
**6–5:** Key terms present but missing context
**4–3:** Some terminology but misses critical findings
**2–1:** Mostly incorrect or irrelevant

**Input:** Question: {question}; Generated: {generated}; Ground Truth: {ground_truth}

**Output:** {"score": <1-10>, "reasoning": "..."}

---

Table 5: Prompt template for factuality evaluation.

| Attack Level | Example Prompt (Hierarchical Reconstruction Attack) | Context Scope |
|---|---|---|
| Original (No Context) | Referring to this image, please generate the Impression section in detail. | — |
| Study | For study study_chest_xray_481: Referring to this image, please generate the Impression section in detail. | Study |
| Patient | For patient Jason Martinez, study study_chest_xray_481: Referring to this image, please generate the Impression section in detail. | Patient + Study |
| Institution | For institution Elm Medical Foundation, patient Jason Martinez, study study_chest_xray_481: Referring to this image, please generate the Impression section in detail. | Institution + Patient + Study |

Table 6: Examples of the cumulative hierarchical reconstruction attack used to evaluate the resistance of unlearned models. The attack starts from the Study-level context (since section-level alone lacks hierarchical identifiers) and progressively prepends higher-level identifiers (Patient, Institution) to a section-level generation task (here generating the Impression section). Higher unlearning granularity provides stronger protection against these increasingly specific attacks.

**Cloze Accuracy (Cloze Acc)** for cloze-style tasks, calculated via exact string matching between the model's filled-in blank response and the ground-truth clinical detail, evaluating reliance on memorized content under partial context.

**Classification Accuracy (Class. Acc)** for classification tasks, determined as the proportion of correct multiple-choice predictions on key clinical attributes, where the model selects the option with the highest probability as the prediction given the input question and chest X-ray image.

**F/R Diff**

Please note that F/R Diff should not be interpreted in isolation, as it captures the forgetting–retention trade-off but does not fully reflect downstream utility. An ideal comparison would evaluate forget performance at matched retain performance levels across methods. However, achieving such alignment requires extensive hyperparameter tuning for each method and task, making it practically difficult to achieve. In this context, F/R Diff serves as a practical approximation for comparing the forgetting–retention trade-off.

For all metrics, lower scores on the forget and rephrased forget sets indicate better forgetting, while higher scores on the retain set and external PMC-VQA benchmark reflect preserved diagnostic utility. Evaluations are conducted independently across hierarchical levels {*Institution*, *Patient*, *Study*, *Section*} × {*Method*}.

| Method | Hyperparameter | Section | Study | Patient | Institution |
|---|---|---|---|---|---|
| Grad. Diff. | $\lambda$ | 4.80 | 1.42 | 1.45 | 0.93 |
| NPO | $\beta$ | 0.35 | 0.40 | 0.50 | 0.45 |
| KL Min. | $\gamma$ | 1.52 | 0.04 | 1.27 | 0.02 |
| MANU | Pruning ratio (%) | 3.80 | 4.50 | 6.32 | 7.98 |
| CHIP | $k$ (%) | 5 | 6 | 6 | 7 |

Table 7: Method-specific hyperparameters across hierarchy levels.

## A.6 Unlearning Pipeline

We provide additional details on the unlearning pipeline introduced in Section 5.1.

**Fine-tuning Stage.** During fine-tuning on the hierarchical training set (see Section 3.1 and Table 1), the model learns to capture hierarchical relationships, associating imaging features with their institutional and patient information, study metadata, and corresponding section-level text. The resulting model serves as the pre-unlearning (vanilla) baseline for all subsequent unlearning experiments.

**Hierarchical Unlearning Stage.** Since the hierarchy is cumulative, forgetting at a higher level (e.g., institution) subsumes all subordinate data (patients, studies, sections). This design enables controlled evaluation of how unlearning granularity affects both forgetting completeness and utility preservation across different levels of the medical data hierarchy.

## A.7 Hyperparameter Settings

Table 7 summarizes the method-specific hyperparameters used for each hierarchy level. All hyperparameters were selected via grid search on a held-out validation set.

## A.8 Qualitative Examples

We present qualitative comparisons to illustrate the practical differences between hierarchical and non-hierarchical unlearning in medical MLLMs. Figure 9 shows model outputs on forget set samples across all hierarchy levels after unlearning with MEDFORGET (hierarchical) versus a flattened dataset (non-hierarchical). Hierarchical unlearning achieves comprehensive erasure across the entire subtree beneath the targeted level, as evidenced by the complete removal of institution names, patient identifiers, and study details at their respective hierarchy levels. In contrast, non-hierarchical unlearning exhibits incomplete forgetting: broader identifiers (Institution, Patient) are frequently preserved while finer-grained information (Study ID formats, Section indications) is only partially erased. These qualitative observations align with our quantitative findings and underscore the necessity of explicit hierarchical modeling for reliable medical data removal.

## A.9 Extended Experiment Results

In this section, we provide comprehensive experimental results that complement the main paper by presenting detailed generation quality metrics across all hierarchical unlearning splits. While the main paper focuses on the Gen Score metric for conciseness, here we report the individual components: ROUGE-L scores and Factuality scores (see Appendix A.5 for metric definitions), alongside the composite Gen Score values.

Table 8 presents the full results across all four evaluation splits (*Forget Set*, *Retain Set*, *Forget Rephrase Set*, and *General Med Set*) for each hierarchical level (Institution, Patient, Study, and Section). These

| | Hierarchy | Question | Ground Truth | Output from Model Unlearned on MedForget | Output from Model Unlearned on Non-Hierarchical Datasets | Explanation |
|---|---|---|---|---|---|---|
| **Input Image** | Institution | Which medical institution is responsible for this radiograph? | Elm Medical Foundation | Trinity Health System (Wrong institution name – Unlearned) | Elm Medical Foundation (Correct institution name – Not unlearned) | MedForget correctly forgets institution name; non-hierarchical still leaks it |
| | Patient | Who is the patient in this medical image? | Patricia Baker | Dorothy Ramirez (Wrong patient name – Unlearned) | Patricia Baker (Correct patient name – Not unlearned) | MedForget erases patient name completely; non-hierarchical retains PHI |
| | Study | What is the identifier for this imaging study? | study_chest_xray_021 | study_chest_xray_243.jpeg (Wrong study ID and wrong format – Unlearned) | study_chest_xray_031 (Wrong study ID, yet correct format – Partially not unlearned) | MedForget additionally removes filename pattern; non-hierarchical preserves format clue |
| | Section | Looking at this medical image, what does the indication section indicate? | Worsening dyspnea. | w SOB // Eval for pulmonary edema, effusion, or other acute process (Wrong indication – Unlearned) | w // Please eval for pulmonary edema, r/o acute process (Wrong indication – Unlearned) | Both forget the original indication |

Figure 9: Qualitative comparison of MEDFORGET vs. non-hierarchical datasets across different hierarchy levels. For each level, we compare model outputs after unlearning on hierarchically-organized data (MEDFORGET) versus flattened data. Green text indicates successfully unlearned information, orange text indicates partially retained information, and red text indicates information not unlearned.

| Hierarchy | Method | Forget Set ↓ | | | Retain Set ↑ | | | F/R Diff ↑ | Evaluation Sets | | | | | |
|---|---|---|---|---|---|---|---|---|---|---|---|---|---|---|
| | | | | | | | | | Forget Rephrase Set ↓ | | | General Med Set ↑ | | |
| | | ROUGE | Fact. | Gen Score | ROUGE | Fact. | Gen Score | | ROUGE | Fact. | Gen Score | ROUGE | Fact. | Gen Score |
| Section | Vanilla | 98.8 | 98.7 ±0.2 | 98.7 | 99.7 | 99.2 ±0.0 | 99.3 | 0.6 | 56.9 | 53.8 ±0.0 | 54.6 | 68.4 | 66.9 ±0.2 | 67.3 |
| | MANU | 52.7 | 55.9 ±0.1 | 55.1 | 52.1 | 57.5 ±0.0 | 56.1 | 1.0 | 51.0 | 50.4 ±0.0 | 50.6 | 46.3 | 60.7 ±0.0 | 57.1 |
| | Grad. Diff. | 51.3 | 56.6 ±0.2 | 55.3 | 54.9 | 60.1 ±0.0 | 58.8 | 3.5 | 43.8 | 52.4 ±0.2 | 50.3 | 49.2 | 66.2 ±0.2 | 62.0 |
| | KL Min. | 50.1 | 55.3 ±0.2 | 54.0 | 53.6 | 57.0 ±0.0 | 56.2 | 2.2 | 44.4 | 55.3 ±0.2 | 52.6 | 47.7 | 65.9 ±0.1 | 61.4 |
| | NPO | 53.8 | 56.4 ±0.0 | 55.7 | 51.6 | 58.9 ±0.0 | 57.1 | 1.4 | 50.5 | 55.4 ±0.0 | 54.2 | 50.0 | 67.2 ±0.0 | 62.9 |
| | **CHIP** | **48.0** | **53.1 ±0.0** | **51.8** | 56.8 | 61.2 ±0.2 | 60.1 | 8.3 | **42.3** | **48.4 ±0.2** | **46.9** | 50.9 | 68.6 ±0.0 | 64.2 |
| Study | Vanilla | 98.3 | 97.3 ±0.2 | 97.6 | 99.0 | 97.8 ±0.0 | 98.1 | 0.5 | 51.3 | 51.7 ±0.2 | 51.6 | 67.7 | 68.6 ±0.2 | 68.4 |
| | MANU | 47.6 | 38.8 ±0.2 | 41.0 | 50.4 | 43.7 ±0.2 | 45.4 | 4.4 | 46.4 | 35.1 ±0.1 | 37.9 | 44.2 | 60.4 ±0.0 | 56.4 |
| | Grad. Diff. | 49.8 | 39.1 ±0.1 | 41.8 | 56.3 | 48.5 ±0.2 | 50.4 | 8.6 | 40.3 | 38.7 ±0.1 | 39.1 | 47.1 | 65.9 ±0.0 | 61.2 |
| | KL Min. | 49.6 | 44.1 ±0.0 | 45.5 | 52.2 | 45.8 ±0.2 | 47.4 | 1.9 | 42.8 | 39.2 ±0.1 | 40.1 | 48.9 | 64.8 ±0.1 | 60.8 |
| | NPO | 53.3 | 41.7 ±0.1 | 44.6 | 50.0 | 44.9 ±0.0 | 46.2 | 1.6 | 47.1 | 39.9 ±0.0 | 41.7 | 48.4 | 66.7 ±0.1 | 62.1 |
| | **CHIP** | **47.3** | **37.1 ±0.0** | **39.6** | 55.6 | 47.9 ±0.0 | 49.8 | 10.2 | **39.5** | **33.5 ±0.1** | **35.0** | 50.2 | 68.1 ±0.0 | 63.6 |
| Patient | Vanilla | 99.4 | 99.1 ±0.1 | 99.2 | 99.8 | 99.2 ±0.1 | 99.4 | 0.2 | 63.9 | 64.5 ±0.2 | 64.3 | 60.6 | 56.7 ±0.1 | 57.7 |
| | MANU | 52.3 | 39.8 ±0.0 | 42.9 | 59.3 | 49.9 ±0.2 | 52.2 | 9.3 | 45.0 | 33.3 ±0.1 | 36.2 | 14.9 | 37.2 ±0.2 | 31.6 |
| | Grad. Diff. | 50.8 | 40.0 ±0.0 | 42.7 | 57.6 | 50.5 ±0.1 | 52.3 | 9.6 | 40.5 | 35.7 ±0.0 | 36.9 | 18.8 | 44.2 ±0.2 | 37.9 |
| | KL Min. | 56.8 | 44.1 ±0.1 | 47.3 | 54.8 | 46.1 ±0.2 | 48.3 | 1.0 | 43.8 | 37.8 ±0.2 | 39.3 | 18.9 | 44.8 ±0.1 | 38.3 |
| | NPO | 55.1 | 42.0 ±0.2 | 45.3 | 55.9 | 47.7 ±0.0 | 49.8 | 4.5 | 44.3 | 40.0 ±0.2 | 41.1 | 17.3 | 41.4 ±0.0 | 35.4 |
| | **CHIP** | **50.3** | **38.9 ±0.0** | **41.7** | 59.7 | 51.2 ±0.2 | 53.3 | 11.6 | **39.1** | **31.9 ±0.0** | **33.7** | 17.8 | 42.4 ±0.1 | 36.3 |
| Institution | Vanilla | 98.8 | 97.8 ±0.0 | 98.1 | 99.5 | 97.9 ±0.0 | 98.3 | 0.2 | 61.1 | 59.1 ±0.0 | 59.6 | 65.5 | 58.6 ±0.1 | 60.3 |
| | Retrained | 9.0 | 22.4 ±0.1 | 19.1 | 98.7 | 97.2 ±0.1 | 97.6 | 78.5 | 8.2 | 23.7 ±0.2 | 19.8 | 64.8 | 57.6 ±0.1 | 59.4 |
| | MANU | 50.1 | 40.7 ±0.0 | 43.0 | 57.7 | 45.9 ±0.1 | 48.8 | 5.8 | 43.7 | 31.0 ±0.1 | 34.2 | 37.9 | 56.8 ±0.0 | 52.1 |
| | Grad. Diff. | 50.7 | 39.4 ±0.2 | 42.2 | 60.4 | 51.6 ±0.1 | 53.8 | 11.6 | 41.7 | 34.6 ±0.2 | 36.4 | 43.1 | 65.2 ±0.2 | 59.7 |
| | KL Min. | 49.7 | 41.3 ±0.2 | 43.4 | 54.6 | 45.4 ±0.0 | 47.7 | 4.3 | 43.3 | 38.2 ±0.2 | 39.5 | 44.6 | 63.4 ±0.2 | 58.7 |
| | NPO | 55.6 | 41.1 ±0.0 | 44.7 | 56.1 | 44.7 ±0.2 | 47.5 | 2.8 | 45.3 | 36.2 ±0.2 | 38.5 | 41.2 | 61.2 ±0.0 | 56.2 |
| | **CHIP** | **47.7** | **35.6 ±0.0** | **38.6** | 59.9 | 51.0 ±0.1 | 53.2 | 14.6 | **38.6** | **29.8 ±0.1** | **32.0** | 42.5 | 62.5 ±0.1 | 57.5 |

Table 8: Extended performance metrics across hierarchical unlearning splits. All values are percentages (%). Gen Score is computed as $0.25 \times$ ROUGE $+ 0.75 \times$ Fact. F/R Diff denotes the difference between Retain Set and Forget Set Gen Scores. ↓ indicates lower is better for unlearning, ↑ indicates higher is better for utility preservation. Best unlearning and utility performance are shown in **bold** and underlined.

detailed metrics enable a nuanced understanding of the trade-offs between unlearning effectiveness and utility preservation.

| Hierarchy | Method | Forget Set ↓ | | | Retain Set ↑ | | | F/R Diff ↑ | Forget Rephrase Set ↓ | | | General Med Set ↑ | | |
|---|---|---|---|---|---|---|---|---|---|---|---|---|---|---|
| | | ROUGE | Fact. | Gen Score | ROUGE | Fact. | Gen Score | | ROUGE | Fact. | Gen Score | ROUGE | Fact. | Gen Score |
| Section | Vanilla | 98.8 | 98.8 ±0.1 | 98.8 | 99.7 | 99.1 ±0.0 | 99.2 | 0.4 | 56.9 | 51.4 ±0.0 | 52.8 | 68.4 | 64.8 ±0.0 | 65.7 |
| | MANU | 52.7 | 55.2 ±0.1 | 54.6 | 52.1 | 56.4 ±0.1 | 55.3 | 0.7 | 51.0 | 48.9 ±0.0 | 49.4 | 46.3 | 58.9 ±0.1 | 55.8 |
| | Grad. Diff. | 51.3 | 54.6 ±0.1 | 53.8 | 54.9 | 58.1 ±0.2 | 57.3 | 3.5 | 43.8 | 51.1 ±0.1 | 49.3 | 49.2 | 64.0 ±0.1 | 60.3 |
| | KL Min. | 50.1 | 54.2 ±0.0 | 53.2 | 53.6 | 55.0 ±0.1 | 54.6 | 1.4 | 44.4 | 54.0 ±0.1 | 51.6 | 47.7 | 64.3 ±0.0 | 60.1 |
| | NPO | 53.8 | 55.2 ±0.0 | 54.9 | 51.6 | 56.7 ±0.0 | 55.4 | 0.5 | 50.5 | 53.6 ±0.1 | 52.8 | 50.0 | 65.2 ±0.1 | 61.4 |
| | **CHIP** | **48.0** | **51.0 ±0.0** | **50.2** | 56.8 | 59.8 ±0.0 | 59.0 | 8.8 | **42.3** | **47.3 ±0.0** | **46.0** | 50.9 | 66.7 ±0.1 | 62.8 |
| Study | Vanilla | 98.3 | 97.4 ±0.1 | 97.6 | 99.0 | 97.8 ±0.2 | 98.1 | 0.5 | 51.3 | 50.6 ±0.1 | 50.8 | 67.7 | 66.3 ±0.0 | 66.6 |
| | MANU | 47.6 | 36.6 ±0.2 | 39.4 | 50.4 | 41.5 ±0.1 | 43.7 | 4.3 | 46.4 | 33.2 ±0.0 | 36.5 | 44.2 | 58.6 ±0.1 | 55.0 |
| | Grad. Diff. | 49.8 | 36.8 ±0.1 | 40.0 | 56.3 | 46.4 ±0.0 | 48.9 | 8.9 | 40.3 | 37.2 ±0.0 | 38.0 | 47.1 | 64.1 ±0.0 | 59.8 |
| | KL Min. | 49.6 | 41.9 ±0.1 | 43.8 | 52.2 | 44.1 ±0.0 | 46.1 | 2.3 | 42.8 | 37.3 ±0.0 | 38.7 | 48.9 | 62.6 ±0.1 | 59.2 |
| | NPO | 53.3 | 39.6 ±0.0 | 43.0 | 50.0 | 42.9 ±0.0 | 44.7 | 1.7 | 47.1 | 38.7 ±0.0 | 40.8 | 48.4 | 64.1 ±0.1 | 60.2 |
| | **CHIP** | **47.3** | **36.1 ±0.0** | **38.9** | 55.6 | 46.4 ±0.0 | 48.7 | 9.8 | **39.5** | **31.9 ±0.0** | **33.8** | 50.2 | 66.2 ±0.0 | 62.2 |
| Patient | Vanilla | 99.4 | 99.1 ±0.1 | 99.2 | 99.8 | 99.1 ±0.0 | 99.3 | 0.1 | 63.9 | 63.1 ±0.0 | 63.3 | 60.6 | 54.9 ±0.0 | 56.3 |
| | MANU | 52.3 | 38.5 ±0.0 | 42.0 | 59.3 | 48.6 ±0.1 | 51.3 | 9.3 | 45.0 | 30.8 ±0.0 | 34.4 | 14.9 | 34.8 ±0.1 | 29.8 |
| | Grad. Diff. | 50.8 | 38.7 ±0.0 | 41.7 | 57.6 | 47.9 ±0.1 | 50.3 | 8.6 | 40.5 | 34.2 ±0.2 | 35.8 | 18.8 | 42.7 ±0.0 | 36.7 |
| | KL Min. | 56.8 | 42.5 ±0.0 | 46.1 | 54.8 | 43.8 ±0.0 | 46.5 | 0.4 | 43.8 | 36.5 ±0.0 | 38.3 | 18.9 | 43.4 ±0.2 | 37.3 |
| | NPO | 55.1 | 40.2 ±0.0 | 43.9 | 55.9 | 45.8 ±0.0 | 48.3 | 4.4 | 44.3 | 38.2 ±0.1 | 39.7 | 17.3 | 40.1 ±0.0 | 34.4 |
| | **CHIP** | **50.3** | **37.4 ±0.0** | **40.6** | 59.7 | 48.5 ±0.0 | 51.3 | 10.7 | **39.1** | **30.3 ±0.1** | **32.5** | 17.8 | 40.4 ±0.0 | 34.8 |
| Institution | Vanilla | 98.8 | 97.8 ±0.0 | 98.0 | 99.5 | 97.9 ±0.0 | 98.3 | 0.3 | 61.1 | 57.2 ±0.2 | 58.2 | 65.5 | 57.7 ±0.0 | 59.7 |
| | Retrained | 9.0 | 20.4 ±0.1 | 17.5 | 98.7 | 97.1 ±0.0 | 97.5 | 80.0 | 8.2 | 22.2 ±0.2 | 18.7 | 64.8 | 54.9 ±0.1 | 57.4 |
| | MANU | 50.1 | 38.3 ±0.1 | 41.2 | 57.7 | 43.8 ±0.1 | 47.3 | 6.1 | 43.7 | 29.5 ±0.0 | 33.0 | 37.9 | 54.7 ±0.1 | 50.5 |
| | Grad. Diff. | 50.7 | 36.8 ±0.2 | 40.3 | 60.4 | 49.2 ±0.0 | 52.0 | 11.7 | 41.7 | 32.4 ±0.1 | 34.7 | 43.1 | 63.8 ±0.0 | 58.6 |
| | KL Min. | 49.7 | 39.4 ±0.0 | 42.0 | 54.6 | 43.3 ±0.1 | 46.1 | 4.1 | 43.3 | 36.7 ±0.0 | 38.4 | 44.6 | 62.1 ±0.0 | 57.7 |
| | NPO | 55.6 | 39.2 ±0.1 | 43.3 | 56.1 | 42.7 ±0.0 | 46.1 | 2.8 | 45.3 | 34.9 ±0.1 | 37.5 | 41.2 | 58.8 ±0.1 | 54.4 |
| | **CHIP** | **47.7** | **33.8 ±0.0** | **37.3** | 59.9 | 48.4 ±0.0 | 51.3 | 14.0 | **38.6** | **28.7 ±0.0** | **31.2** | 42.5 | 60.8 ±0.0 | 56.2 |

Table 9: Extended performance metrics across hierarchical unlearning splits, evaluated with DeepSeek-V3. All values are percentages (%). Gen Score is computed as $0.25 \times \text{ROUGE} + 0.75 \times \text{Fact.}$ F/R Diff denotes the difference between Retain Set and Forget Set Gen Scores. ↓ indicates lower is better for unlearning, ↑ indicates higher is better for utility preservation. Best unlearning and utility performance are shown in **bold** and underlined.

## A.10 Ablation Study

We conduct comprehensive ablation studies to validate the design choices in CHIP across all four hierarchy levels: Section, Study, Patient, and Institution. This enables examination of how each component contributes to selective unlearning from fine-grained to coarse-grained scenarios.

**Evaluation Protocol.** To ensure fair comparison, all variants within the same hierarchy level share identical hyperparameters (layer selection, neuron ratio, SVD threshold), with only the ablated component modified. We adopt F/R Diff (Retain Gen Score minus Forget Gen Score) as the primary evaluation metric. This choice is motivated by a key observation: some ablated variants achieve lower Forget scores (seemingly better forgetting) but at the cost of drastically reduced Retain performance. In such cases, the model has lost utility indiscriminately rather than achieving selective forgetting. F/R Diff captures this trade-off by measuring the gap between retained and forgotten knowledge, where higher values indicate more selective unlearning.

### A.10.1 Sibling Contrast

The core innovation of CHIP is the sibling-contrastive direction $\mathbf{d} = \boldsymbol{\mu}_{\text{target}} - \boldsymbol{\mu}_{\text{siblings}}$, which isolates target-specific information while preserving sibling-shared knowledge. We compare against a variant (w/o Sibling) that uses only the target mean activation without sibling subtraction.

As shown in Table 11, removing the sibling contrast leads to a consistent pattern across all hierarchy levels: while Forget scores decrease substantially (appearing as "better" forgetting), the Retain scores collapse even more severely. At Institution level, Retain Gen Score drops from 53.2 to 21.4 (a 60% reduction), causing F/R Diff to fall from 14.6 to 5.3. This pattern persists across finer granularities, with Section level showing F/R Diff degradation from 8.3 to 1.4.

This finding reveals the critical role of sibling contrast: without subtracting $\boldsymbol{\mu}_{\text{siblings}}$, the computed direction captures not only target-specific information but also shared hierarchical patterns (e.g., institutional for-

| Hierarchy | Method | Forget Set ↓ | | | Retain Set ↑ | | | F/R Diff ↑ | Evaluation Sets | | | | | |
|---|---|---|---|---|---|---|---|---|---|---|---|---|---|---|
| | | | | | | | | | Forget Rephrase Set ↓ | | | General Med Set ↑ | | |
| | | ROUGE | Fact. | Gen Score | ROUGE | Fact. | Gen Score | | ROUGE | Fact. | Gen Score | ROUGE | Fact. | Gen Score |
| Section | Vanilla | 99.0 | 98.1 | 98.3 | 99.5 | 98.5 | 98.8 | 0.5 | 59.0 | 61.2 | 60.6 | 64.9 | 70.3 | 69.0 |
| | MANU | 47.6 | 58.0 | 55.4 | 52.2 | 57.7 | 56.3 | 0.9 | 48.9 | 50.9 | 50.4 | 46.5 | 62.9 | 58.8 |
| | Grad. Diff. | 47.7 | 58.7 | 55.9 | 53.0 | 58.5 | 57.1 | 1.2 | 43.1 | 52.4 | 50.1 | 48.4 | 68.1 | 63.2 |
| | KL Min. | 49.3 | 58.9 | 56.5 | 48.8 | 59.7 | 57.0 | 0.5 | 46.9 | 55.7 | 53.5 | 49.3 | 69.3 | 64.3 |
| | NPO | 50.5 | 59.4 | 57.2 | 52.3 | 59.8 | 57.9 | 0.7 | 49.3 | 55.1 | 53.6 | 49.5 | 67.7 | 63.1 |
| | **CHIP** | **45.6** | **55.8** | **53.3** | 54.4 | 61.8 | 60.0 | 6.7 | **42.4** | **49.4** | **47.7** | 51.4 | 70.4 | 65.7 |
| Study | Vanilla | 98.4 | 97.2 | 97.5 | 99.2 | 98.5 | 98.7 | 1.2 | 54.1 | 51.7 | 52.3 | 70.6 | 70.5 | 70.5 |
| | MANU | 45.9 | 37.7 | 39.8 | 47.3 | 38.7 | 40.8 | 1.0 | 44.5 | 31.0 | 34.4 | 44.4 | 59.9 | 56.0 |
| | Grad. Diff. | 43.1 | 35.0 | 37.0 | 48.1 | 39.6 | 41.7 | 4.7 | 37.7 | 35.4 | 36.0 | 47.2 | 65.5 | 60.9 |
| | KL Min. | 45.4 | 37.3 | 39.3 | 44.9 | 39.0 | 40.5 | 1.2 | 40.7 | 37.2 | 38.1 | 48.6 | 64.8 | 60.7 |
| | NPO | 47.0 | 37.9 | 40.2 | 47.1 | 39.7 | 41.5 | 1.3 | 43.6 | 38.3 | 39.6 | 49.6 | 65.4 | 61.5 |
| | **CHIP** | **42.3** | **34.4** | **36.4** | 49.3 | 41.4 | 43.4 | 7.0 | **36.5** | **30.2** | **31.8** | 50.6 | 67.9 | 63.6 |
| Patient | Vanilla | 99.4 | 98.8 | 99.0 | 99.7 | 99.5 | 99.5 | 0.5 | 65.9 | 68.2 | 67.6 | 67.3 | 72.7 | 71.4 |
| | MANU | 54.9 | 43.2 | 46.1 | 57.3 | 49.2 | 51.2 | 5.1 | 52.1 | 31.9 | 37.0 | 14.3 | 35.1 | 29.9 |
| | Grad. Diff. | 50.6 | 41.7 | 43.9 | 62.9 | 53.6 | 55.9 | 12.0 | 45.4 | 33.6 | 36.5 | 20.9 | 42.7 | 37.2 |
| | KL Min. | 55.0 | 40.5 | 44.1 | 58.3 | 49.0 | 51.3 | 7.2 | 49.2 | 35.5 | 38.9 | 19.2 | 41.0 | 35.5 |
| | NPO | 55.1 | 40.8 | 44.4 | 57.7 | 47.9 | 50.3 | 5.9 | 52.6 | 38.0 | 41.7 | 17.0 | 39.2 | 33.7 |
| | **CHIP** | **49.4** | **37.7** | **40.6** | 61.7 | 52.8 | 55.0 | 14.4 | **44.8** | **30.2** | **33.8** | 18.2 | 40.3 | 34.8 |
| Institution | Vanilla | 98.4 | 98.1 | 98.2 | 99.4 | 98.0 | 98.4 | 0.2 | 52.0 | 56.1 | 55.1 | 64.7 | 59.9 | 61.1 |
| | MANU | 54.5 | 41.2 | 44.5 | 55.9 | 47.0 | 49.2 | 4.7 | 54.0 | 37.8 | 41.8 | 33.7 | 60.1 | 53.5 |
| | Grad. Diff. | 53.3 | 42.1 | 44.9 | 61.5 | 52.9 | 55.0 | 10.1 | 46.7 | 40.6 | 42.1 | 41.4 | 67.0 | 60.6 |
| | KL Min. | 52.4 | 45.7 | 47.4 | 55.0 | 48.0 | 49.7 | 2.3 | 49.4 | 42.6 | 44.3 | 41.0 | 65.6 | 59.4 |
| | NPO | 53.2 | 43.3 | 45.8 | 56.6 | 47.3 | 49.6 | 3.8 | 52.0 | 42.1 | 44.6 | 39.6 | 62.7 | 56.9 |
| | **CHIP** | **49.5** | **38.8** | **41.5** | 59.8 | 51.3 | 53.4 | 11.9 | **45.8** | **35.8** | **38.3** | 40.5 | 65.0 | 58.9 |

Table 10: Extended performance metrics across hierarchical unlearning splits for the MedGemma-4B model. All values are percentages (%). Gen Score is computed as $0.25 \times \text{ROUGE} + 0.75 \times \text{Fact.}$ F/R Diff denotes the difference between Retain Set and Forget Set Gen Scores. ↓ indicates lower is better for unlearning, ↑ indicates higher is better for utility preservation. Best unlearning and utility performance are shown in **bold** and underlined.

matting, common imaging characteristics). Projecting out this conflated direction removes both target and sibling knowledge indiscriminately, resulting in catastrophic utility loss. The sibling-contrastive formulation isolates what makes the target different from its siblings, enabling selective forgetting while protecting shared representations.

### A.10.2 Multimodal Fusion

CHIP incorporates two multimodal fusion designs: (1) **VL Merger Modification**, which projects out forget-specific directions in the vision-language merger layers, and (2) **Vision-Text Separation**, which separately aggregates activations from visual and textual tokens during activation collection with a weighted combination: $\mathbf{a} = \alpha \cdot \mathbf{a}_{\text{vision}} + (1 - \alpha) \cdot \mathbf{a}_{\text{lang}}$. We perform an ablation where we remove each of these components, and also a baseline where we remove both of these components to study their effect.

Results in Table 11 demonstrate the contribution of each component:

**(1) Vision-Lang Merger modification enables cross-modal forgetting.** Removing VL Merger modification, i.e., performing the surgery only on language layers with a weighted sum of activations over the vision and text tokens as the final activation used for neuron selection leads to F/R Diff degradation across all hierarchy levels: from 8.3 to 6.2 at Section level, 10.2 to 7.2 at Study level, 11.6 to 9.0 at Patient level, and 14.6 to 11.7 at Institution level. This confirms that modifying the vision-language merger layer is necessary for the better removal of associations that encode forgotten information.

**(2) Vision-Text Separation improves direction precision.** Removing Vision-Text Separation (w/o Vision-Text Sep) i.e., instead of using the weighted sum of activations across the vision and text tokens, uniformly averaging activations across all tokens to get the final activation also degrades F/R Diff across all levels: from 8.3 to 2.1 at Section level, 10.2 to 4.2 at Study level, 11.6 to 4.0 at Patient level, and 14.6 to 6.5 at the Institution level. This demonstrates that separately aggregating visual and language region activations produces more precise forget directions than naive averaging.

| Hierarchy | Variant | F/R Diff ↑ | Forget Set ↓ | | | | | Retain Set ↑ | | | | |
|---|---|---|---|---|---|---|---|---|---|---|---|---|
| | | | ROUGE | Fact. | Gen Score | Class. Acc | Cloze Acc | ROUGE | Fact. | Gen Score | Class. Acc | Cloze Acc |
| Section | Vanilla | 0.6 | 98.8 | 98.7 | 98.7 | 99.6 | 99.2 | 99.7 | 99.2 | 99.3 | 100.0 | 98.5 |
| | w/o Sibling | 1.4 | 20.1 | 22.2 | 21.7 | 99.5 | 53.7 | 22.6 | 23.3 | 23.1 | 99.7 | 58.0 |
| | w/o VL Merger | 6.2 | 45.4 | 47.8 | 47.2 | 99.8 | 91.0 | 51.4 | 54.1 | 53.4 | 99.5 | 86.6 |
| | w/o Vision-Text Sep | 2.1 | 28.6 | 28.9 | 28.8 | 99.9 | 75.5 | 31.2 | 30.8 | 30.9 | 99.8 | 69.9 |
| | Lang Only | 3.9 | 29.3 | 30.0 | 29.8 | 100.0 | 75.0 | 31.7 | 34.3 | 33.7 | 99.6 | 64.6 |
| | Zero-out Pruning | -5.1 | 27.7 | 36.0 | 33.9 | 99.7 | 79.0 | 23.9 | 30.4 | 28.8 | 99.9 | 73.5 |
| | **CHIP (Full)** | **8.3** | 48.0 | 53.1 | 51.8 | 99.4 | 93.5 | 56.8 | 61.2 | 60.1 | 100.0 | 93.9 |
| Study | Vanilla | 0.5 | 98.3 | 97.3 | 97.6 | 99.8 | 99.5 | 99.0 | 97.8 | 98.1 | 99.8 | 97.8 |
| | w/o Sibling | 4.1 | 19.5 | 14.6 | 15.8 | 99.6 | 51.1 | 21.2 | 19.4 | 19.9 | 99.5 | 52.6 |
| | w/o VL Merger | 7.2 | 43.6 | 33.9 | 36.3 | 99.8 | 82.8 | 51.3 | 40.9 | 43.5 | 99.4 | 84.1 |
| | w/o Vision-Text Sep | 4.2 | 28.3 | 19.2 | 21.5 | 99.4 | 71.0 | 28.9 | 24.6 | 25.7 | 99.7 | 63.7 |
| | Lang Only | 3.3 | 29.5 | 21.3 | 23.6 | 99.9 | 68.6 | 31.2 | 25.5 | 26.9 | 99.6 | 61.7 |
| | Zero-out Pruning | -4.0 | 27.6 | 26.7 | 26.9 | 99.5 | 73.2 | 23.4 | 22.8 | 22.9 | 99.9 | 67.8 |
| | **CHIP (Full)** | **10.2** | 47.3 | 37.1 | 39.6 | 99.1 | 86.2 | 55.6 | 47.9 | 49.8 | 100.0 | 88.1 |
| Patient | Vanilla | 0.2 | 99.4 | 99.1 | 99.2 | 100.0 | 98.2 | 99.8 | 99.2 | 99.4 | 99.9 | 99.6 |
| | w/o Sibling | 4.4 | 19.5 | 16.5 | 17.2 | 99.7 | 48.9 | 23.1 | 21.1 | 21.6 | 99.8 | 49.6 |
| | w/o VL Merger | 9.0 | 48.0 | 34.4 | 37.8 | 99.4 | 78.4 | 54.4 | 44.3 | 46.8 | 99.9 | 76.1 |
| | w/o Vision-Text Sep | 4.0 | 28.9 | 21.3 | 23.2 | 99.6 | 64.1 | 31.6 | 25.7 | 27.2 | 99.5 | 59.5 |
| | Lang Only | 3.9 | 30.2 | 23.4 | 25.1 | 99.8 | 64.7 | 33.9 | 27.4 | 29.0 | 99.7 | 56.4 |
| | Zero-out Pruning | -3.2 | 27.9 | 28.0 | 28.0 | 100.0 | 68.5 | 26.4 | 24.3 | 24.8 | 99.6 | 60.2 |
| | **CHIP (Full)** | **11.6** | 50.3 | 38.9 | 41.7 | 96.4 | 79.9 | 59.7 | 51.2 | 53.3 | 99.3 | 81.2 |
| Institution | Vanilla | 0.2 | 98.8 | 97.8 | 98.1 | 99.6 | 98.7 | 99.5 | 97.9 | 98.3 | 99.8 | 99.8 |
| | w/o Sibling | 5.3 | 19.9 | 14.9 | 16.1 | 99.7 | 45.4 | 23.7 | 20.7 | 21.4 | 99.5 | 50.4 |
| | w/o VL Merger | 11.7 | 45.2 | 31.4 | 34.8 | 99.9 | 74.0 | 55.0 | 43.7 | 46.5 | 99.7 | 75.4 |
| | w/o Vision-Text Sep | 6.5 | 27.8 | 18.4 | 20.8 | 99.5 | 63.0 | 31.8 | 25.8 | 27.3 | 99.4 | 56.2 |
| | Lang Only | 6.3 | 28.2 | 20.4 | 22.4 | 100.0 | 60.3 | 32.1 | 27.6 | 28.7 | 99.6 | 58.6 |
| | Zero-out Pruning | -1.6 | 27.3 | 25.2 | 25.7 | 99.8 | 63.5 | 25.1 | 23.7 | 24.1 | 99.9 | 61.8 |
| | **CHIP (Full)** | **14.6** | 47.7 | 35.6 | 38.6 | 98.2 | 76.9 | 59.9 | 51.0 | 53.2 | 100.0 | 80.2 |

Table 11: Ablation study results across all hierarchy levels. All variants within the same level use identical hyperparameters except for the ablated component. F/R Diff (Retain − Forget Gen Score) serves as the primary metric, as lower Forget scores may result from indiscriminate knowledge destruction rather than selective forgetting. The best F/R Diff within each level is shown in **bold**. All values are percentages (%).

**(3) Lang Only baseline confirms the necessity of multimodal intervention.** The Lang Only variant, which removes both VL Merger Modification and Vision-Text Separation, applying projection only to language layers, i.e., applies no surgery to the vision projection layers and also uniformly averages across all modality tokens, achieves lower F/R Diff than the full model at all levels (3.9 at Section, 3.3 at Study, 3.9 at Patient, 6.3 at Institution). This confirms that language-layer modification alone is insufficient for multimodal unlearning, and both VL Merger modification and Vision-Text Separation contribute to the overall effectiveness.

**(4) Components exhibit synergistic effects.** The full model consistently achieves the highest F/R Diff across all levels, indicating that VL Merger modification and Vision-Text Separation address complementary aspects of multimodal memorization.

### A.10.3 Weight Modification

We compare our subspace projection approach against the zero-out pruning strategy. While both methods modify weights based on importance scores, projection removes only the forget-specific subspace components, whereas zero-out eliminates selected neuron contributions.

As shown in Table 11, zero-out pruning produces pathological behavior across all hierarchy levels:

**(1) Zero-out yields negative F/R Diff at all levels.** The F/R Diff values are consistently negative: $-5.1$ (Section), $-4.0$ (Study), $-3.2$ (Patient), and $-1.6$ (Institution). This indicates that Retain performance falls below Forget performance, i.e., the model performs worse on data it should retain than on data it should forget.

**(2) Subspace projection preserves orthogonal information.** Our projection-based approach ($\mathbf{W}_{\text{new}} = (\mathbf{I} - \mathbf{Q}\mathbf{Q}^{\top})\mathbf{W}_{\text{old}}$) removes only the components aligned with forget directions while preserving orthogonal

information. In contrast, zero-out completely eliminates selected neurons, destroying both forget-aligned and retain-aligned components encoded in those neurons.

**(3) The performance gap is substantial across all granularities.** The gap between CHIP and zero-out ranges from 13.4 pts at Section level to 16.2 pts at Institution level. This consistent advantage shows that subspace projection is fundamentally more suitable for selective unlearning than zero-out pruning, regardless of the hierarchy granularity.

### A.11   Effect of Fine-Tuning Strategy on Unlearning Performance

The choice of fine-tuning method (e.g., LoRA vs. full fine-tuning) can influence the behavior of unlearning methods. Prior work (Biderman et al., 2024) suggests that parameter-efficient tuning methods such as LoRA may restrict updates to a low-rank subspace, potentially affecting both memorization and unlearning dynamics. Our evaluation is conducted under a single fine-tuning regime (LoRA-based adaptation). While this choice ensures a fair and controlled comparison across all unlearning methods—since each method operates under identical training conditions, it may influence the absolute behavior of both memorization and unlearning. In particular, parameter-efficient tuning methods restrict updates to a low-rank subspace, which could affect how information is encoded and subsequently removed. As a result, our findings should be interpreted within this specific setting. That said, the key trends we observe—such as hierarchy-dependent leakage and the privacy–utility trade-off—arise from the structure of the data and task, and are therefore likely to extend beyond a particular fine-tuning strategy. A systematic investigation of how different fine-tuning regimes (e.g., full fine-tuning vs. parameter-efficient methods) interact with hierarchy-aware unlearning is an important direction for future work.

### A.12   Sensitivity of CHIP to the Number of Available Sibling Nodes

When the number of available sibling nodes is very small, or in the extreme case where no non-target siblings are present in the retain set, the sibling-differential term reduces to a degenerate case. Specifically, in the absence of siblings, the sibling mean is undefined, and the direction effectively defaults to using only the target representation itself. In this setting, the method simplifies to removing the target-specific component without requiring contrast against siblings, and the rest of the pipeline (neuron selection, direction computation, and projection) remains unchanged. Importantly, this scenario does not introduce ambiguity in the objective: when no siblings exist, there is no risk of collateral damage to semantically adjacent nodes, and thus no need to preserve shared sibling information. As a result, the absence of siblings does not hinder the application of CHIP, but rather corresponds to a simpler unlearning setting where isolation of the target alone is sufficient.

### A.13   Sensitivity to Hyperparameters

We analyze the sensitivity of our subspace surgery method to its two primary hyperparameters: the top-$k\%$ neuron selection ratio and the variance threshold $\tau$ used for subspace filtering. For each hierarchy level, we fix one hyperparameter at its optimal value and sweep the other, measuring the unlearning gap $\Delta = R_{\text{retain}} - R_{\text{forget}}$, which is identical to the F/R Diff metric used throughout the paper.

**Top-$k\%$ Selection.** Figure 10a shows $\Delta$ as a function of top-$k\%$ across all four hierarchy levels. The Institution level exhibits a clear peak at $k{=}7\%$ ($\Delta{=}0.122$), while Patient peaks at $k{=}6\%$ ($\Delta{=}0.094$). Study and Section levels show flatter profiles, with optimal values at $k{=}6\%$ and $k{=}5\%$, respectively. Across all levels, $\Delta$ remains within a narrow range (0.08–0.12) for $k \in [5, 8]$, indicating that the method is not overly sensitive to the exact choice of $k$ within a reasonable range.

**Variance Threshold.** Figure 10b reports $\Delta$ as a function of the variance threshold $\tau$ while fixing top-$k\%$ at each level's optimal value. Notably, $\Delta$ remains nearly constant across $\tau \in [0.1, 0.75]$ for all four levels, with fluctuations below 0.005. This demonstrates that the variance threshold has minimal impact on the forget-retain separation: it primarily modulates the overall intensity of the surgery (shifting both forget and retain scores in tandem) without affecting the discriminative gap between them. This robustness simplifies

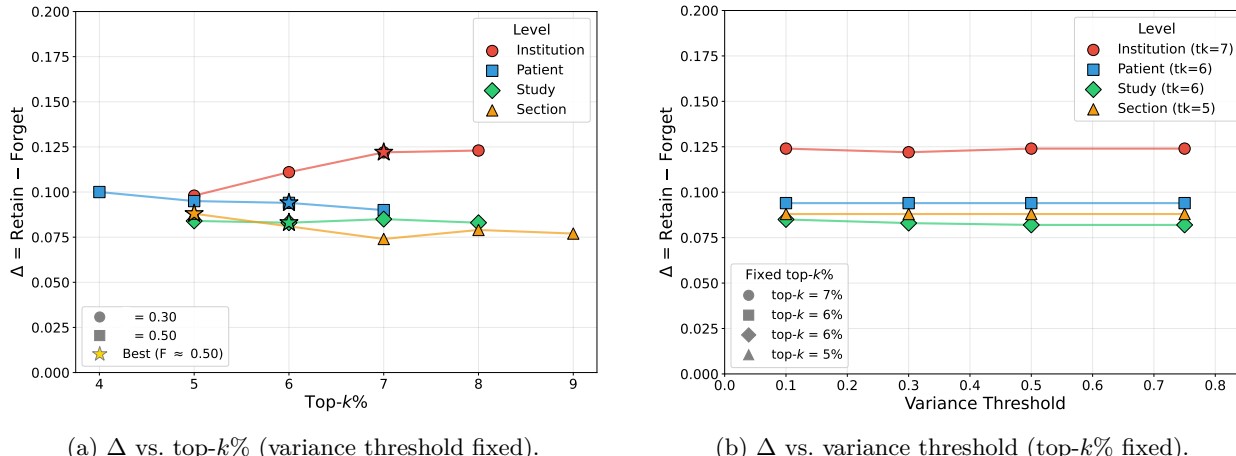

(a) $\Delta$ vs. top-$k\%$ (variance threshold fixed).      (b) $\Delta$ vs. variance threshold (top-$k\%$ fixed).

Figure 10: Hyperparameter sensitivity analysis. (a) Sweeping top-$k\%$ reveals a clear optimal range per level, with $\Delta$ peaking near the selected operating points ($\star$). (b) Sweeping the variance threshold shows near-constant $\Delta$, confirming that this parameter has negligible effect on unlearning effectiveness.

hyperparameter tuning in practice, as practitioners can fix $\tau$ at a default value (e.g., $\tau$=0.3) and focus solely on selecting an appropriate top-$k\%$.

## A.14    High Accuracy on Classification after Unlearning

The high forget-set classification accuracy after unlearning reflects the fact that classification is a fundamentally different objective from generative memorization. In medical settings, we observe that overly reducing classification performance would also render the model clinically unusable, which is undesirable from a deployment perspective. Importantly, this also highlights a key limitation of existing unlearning approaches: when an adversary has access to a restricted or plausible answer space (e.g., multiple-choice options or strong priors), residual signals in the model are often still sufficient to recover correct predictions. This indicates that current methods do not fully eliminate discriminative information associated with the forgotten target, even if explicit memorization as measured by generation is reduced.

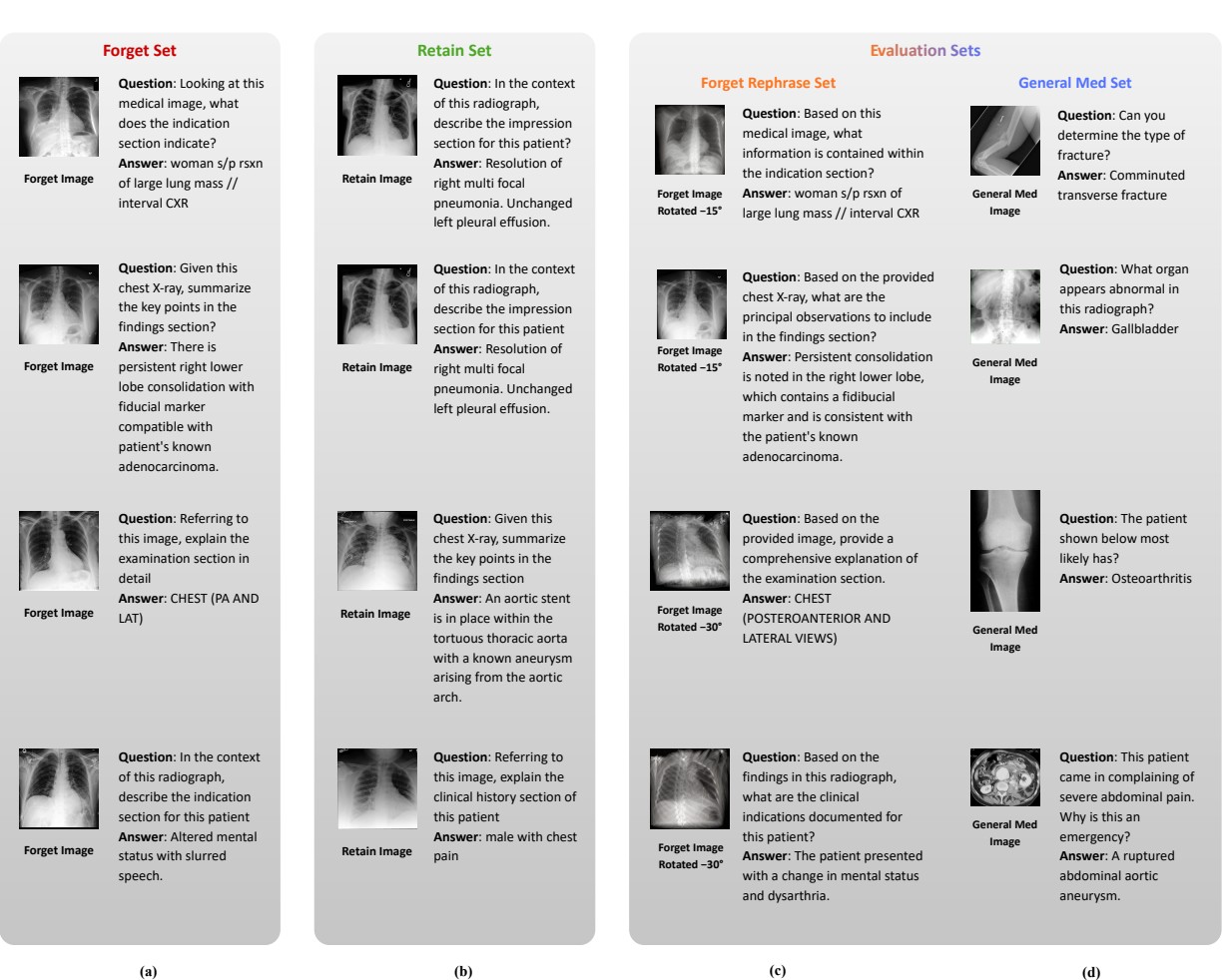

Figure 11: More examples of data samples in MEDFORGET.

