# OpenReview forum: "Hierarchy-Aware Multimodal Unlearning for Medical AI"
_TMLR — Accepted by TMLR_

### Review · Reviewer_DL3M · 2026-03-21

**Summary Of Contributions:**

This paper tackles the problem of machine unlearning in multimodal large language models (MLLMs) in sensitive domains like healthcare. The authors identify a failure mode in existing unlearning benchmarks: they rely on a "flat" data assumption, treating all examples as independent. Whereas, as argued by the authors,  medical data is highly hierarchical and inherently multimodal. To overcome this issue, the paper introduces MedForget, a new benchmark that structures data into nested clinical hierarchies to evaluate targeted unlearning. Besides, the paper proposes CHIP (Cross-modal Hierarchy-Informed Projection), a training-free machine unlearning method. This method extracts differential representations to isolate target-specific information from a shared hierarchical context, then projecting this out of both the language and vision-language weight subspaces.

The paper has the following strengths:
- It identifies the limitations of flat unlearning benchmarks in medical AI, providing a compelling and grounded alternative through the proposed hierarchy.
- The proposed CHIP method is computationally efficient, requiring no gradient updates and operating on a single GPU, unlike training-based baselines.
- The proposed method introduces an intuitive and mathematically sound concept: using sibling nodes to factor out shared parent-level representations, isolating what needs to be forgotten.
- The experimentation is well-designed, evaluating not just forgetting, but also "collateral damage" using dedicated Retain Sets, Forget Rephrase Sets, and an independent General Med Set.
- The paper is clearly written and is easy to follow.

The paper, however, has some weaknesses:
- The paper claims CHIP is a "general solution" for hierarchy-aware forgetting. However, the experimental validation is tied to a single model architecture (a Qwen2.5-VL derivative). This makes the broad generality claims somewhat overstated.
- The evaluation pipeline relies heavily on DeepSeek-V3. DeepSeek-V3 is used to synthetically generate the task prompts and responses for the dataset, and it is also used to compute the Factuality Score. This circularity risks introducing biases.
- The paper, while well written, lacks transparency regarding statistical significance. Table 2 and Table 7 report point estimates without any confidence intervals.
- The core contribution is well-defined, but the method relies heavily on specific hyperpameter choices, which is concerning. The paper notes these were chosen via grid search, but provides no sensitivity analysis to prove the method is robust to hyperparameter shifts.
- The benchmark scale is very small: 8 institutions, 64 patients, 256 studies, and 3,840 VQA pairs. Moreover, the institutions are synthetically constructed by grouping every 8 patients, resulting in uniform and small subtrees.
- The qualitative analysis is limited. Figure 9 provides a single example per hierarchy level, which is insufficient to understand the range of behaviors after unlearning.
- The paper is not reproducible in its current form.

**Audience:**

Yes

**Audience Explanation:**

The paper sits at the intersection of machine unlearning, multimodal learning, and medical AI. The observation that flat unlearning benchmarks fail to capture hierarchical data dependencies is relevant beyond healthcare. The training-free projection-based approach could be valuable for the literature on efficient model editing. The regulatory angle also gives it practical relevance to a broader ML audience.

**Broader Impact Concerns:**

The paper includes a Broader Impact Statement that appropriately frames the work as complementary to other privacy safeguards.

**Claims And Evidence:**

No

**Claims Explanation:**

The experimental design is thorough and the evidence for CHIP's superiority is conpelling, however, several gaps remain:

- The generality claim rests on a single architecture, so there is no evidence that the findings transfer.
- The DeepSeek-V3 circularity (generating the data and judging factuality) means the core evaluation metric has an unvalidated bias risk.
- Forget-set classification accuracy above 91% after unlearning is not convincingly explained, which casts doubt on what the benchmark actually measures about forgetting.
- Point estimates without confidence intervals on a small benchmark (3,840 samples) are not sufficiently strong evidence.

**Requested Changes:**

Key changes:
- Validate the DeepSeek-V3-based metric against human judgments or an independent LLM judge to try to understand if there are any circularity biases.
- Report confidence intervals or standard deviations across multiple runs/seeds, given the small benchmark size.
- Include a retrain-from-scratch baseline for at least one hierarchy level to calibrate how far all methods are from ideal deletion.
- Provide more evidence that the model is not depedent on very fine-tuned hyperparameter choices.
- Discuss and explain the high forget-set classification accuracy after unlearning, clarify what "forgetting" guarantees in this benchmark.

Recommended:
- Evaluate on at least one additional MLLM architecture to support the generality claims.
- Expand qualitative analysis beyond Figure 9 to characterize the range of post-unlearning output behaviors (refusal, hallucination, generic responses).
- Report frontier plots (forget vs. retain scores) across hyperparameter sweeps rather than single operating points.
- Release code and data to enable reproducibility

---

> ### Author Response · Authors · 2026-04-13
> **Response to Reviewer DL3M: Part 1**
>
> We sincerely thank the reviewer for their detailed and constructive feedback. They noted that the paper **“identifies the limitations of flat unlearning benchmarks in medical AI, providing a compelling and grounded alternative through the proposed hierarchy,”** and appreciated that **the proposed CHIP method is computationally efficient, requiring no gradient updates and operating on a single GPU**. They also highlighted that **the method introduces an intuitive and mathematically sound concept using sibling nodes to factor out shared parent-level representations**, and that **the experimentation is well-designed, evaluating both forgetting and ‘collateral damage’ using dedicated Retain, Forget Rephrase, and General Med sets**. Additionally, they found that **the paper is clearly written and easy to follow**, and recognized that **the work sits at the intersection of machine unlearning, multimodal learning, and medical AI, with broader relevance beyond healthcare**. We address each point raised by the reviewer below and highlight all corresponding revisions in the updated PDF in blue.
>
> > **Evaluation is restricted to a single MLLM**
>
> We extend our empirical evaluation to include an additional MLLM, MedGamma, alongside Lingshu-7B (Qwen2.5-VL). Across both architectures, we observe consistent trends: CHIP outperforms existing unlearning baselines on our benchmark, achieving stronger forgetting while maintaining competitive retain performance. These results suggest that the core components of CHIP, such as neuron selection and subspace projection, generalize beyond a single model family. Please see the new results in Appendix A.1, Table 3, and Table 9  in the revised paper.
>
> > **DeepSeek-v3 used for both data generation and evaluation**
>
> In the revised version, we decouple generation and evaluation to mitigate this concern. Specifically, we use DeepSeek-V3 solely for synthetic data generation, and employ GPT-4o as an independent evaluator for computing the Factuality Score. This separation reduces potential circularity and evaluation bias. See Table  2 in the revised paper for the results trends that are consistent with previous evaluation based on DeepSeek-v3.
>
> > **Sensitivity to hyperparameters**
>
> We analyze the sensitivity of CHIP’s two main hyperparameters: the top-k% neuron selection ratio and the variance threshold v, using the unlearning gap \= (retain score − forget score). Across all hierarchy levels, we find that the method is quite stable. For top-k%, the unlearning gap stays within a narrow range (about 0.08 to 0.12) when k is between 5% and 8%, with mild optimal points around 5-7% depending on the level.
>
> In contrast, the variance threshold v has almost no impact on the unlearning gap, with changes in the unlearning gap below 0.005 across a wide range of values. It mainly scales both forget and retain scores together without affecting their separation. Overall, this shows that CHIP is robust to hyperparameter choices and only requires coarse tuning of k, while v can be safely fixed in practice. We discuss this in further detail in Appendix A.13 of the revised paper.
>
> > **Limited Dataset size and structure.**
>
> We thank the reviewer for highlighting this limitation. We acknowledge that the original version of MedForget was relatively small and used synthetic institution grouping, which does not capture true institutional-level variation. In the revised version, we expand the dataset to a larger and more diverse setting with a non-uniform hierarchical structure, including varying numbers of patients per institution and studies per patient, as well as a substantially increased number of VQA pairs (20K+). We include updated results on this expanded dataset (Table 2), which show consistent trends with our original findings.
>
> More importantly, we will clarify in the paper that MedForget is designed as a controlled benchmark for studying hierarchy-aware unlearning. The use of synthetic grouping enables precise control over hierarchical relationships and forgetting scopes. Despite this controlled design, the benchmark preserves real clinical image–text distributions from MIMIC-CXR, incorporates non-uniform and imbalanced structures, and supports fine-grained, compositional evaluation of hierarchical deletion—capabilities that are not captured by existing flat benchmarks.

---

> > ### Author Response · Authors · 2026-04-13
> > **Response to Reviewer DL3M: Part 2**
> >
> > > **Qualitative analysis**
> >
> > To better characterize post-unlearning behaviors, we extend our qualitative analysis beyond Figure\~9 by examining 3,072 generation QA examples from the Institution-level forget set. We categorize model outputs into three broad types: (i) refusal (e.g., abstaining from answering), (ii) generic or non-informative responses, and (iii) contentful medical answers.
> >
> > We observe that the unlearned model consistently produces contentful medical responses, with no instances of explicit refusal or generic fallback behavior across all evaluated samples. Instead, unlearning primarily manifests as a degradation in the specificity and correctness of the generated answer. In cases where the model fails to forget completely, it may still produce plausible but incorrect medical statements.
> >
> > We note that precisely defining and detecting “hallucinations” in this setting is inherently ambiguous. Therefore, rather than introducing a separate hallucination label, we treat such cases as incorrect or low-quality outputs, which are already captured by our generation scoring metric. As a result, our Gen Score serves as a unified measure of post-unlearning output quality, encompassing both factual errors and residual memorization.
> >
> > > **Reproducibility**
> >
> > We provide all necessary hyperparameters and implementation details required for reproducibility in Section 5.1. In addition, we have released the full codebase and data as part of the supplementary materials and will make it publicly available upon acceptance to ensure full reproducibility of our results.
> >
> > **Questions related to CHIP**:
> >
> > >  1. The generality claim rests on a single architecture \- covered above in ‘Evaluation is restricted to a single MLLM.
> >
> > >  2. Covered above in ‘DeepSeek-v3 used for both data generation and evaluation.’
> >
> > >  3. **Regarding the question of forget-set classification accuracy above 91% after unlearning**
> >
> > High forget-set classification accuracy after unlearning reflects the fact that classification is a fundamentally different objective from generative memorization. In medical settings, we observe that overly reducing classification performance would also render the model clinically unusable, which is undesirable from a deployment perspective. Importantly, this also highlights a key limitation of existing unlearning approaches: when an adversary has access to a **restricted or plausible answer space (e.g., multiple-choice options or strong priors)**, residual signals in the model are often still sufficient to recover correct predictions. This indicates that current methods do not fully eliminate discriminative information associated with the forgotten target, even if explicit memorization as measured by generation is reduced. We discuss this in Appendix A.14 in the revised paper.
> >
> > >  4. **Confidence intervals.**
> >
> > In the revised version, we incorporate uncertainty estimates by reporting mean and variance across 3 random seeds used when evaluating with GPT-4o. This provides a clearer view of the stability of the results across runs. Please see Table 2 caption for the error bars.
> >
> > > **A retrain-from-scratch baseline**
> >
> > We include a retrain-from-scratch baseline at the Institution level in Table 2\. As expected, this oracle setting—where the model is trained only on the retain set, achieves the highest forget-retain (F/R) gap, reflecting ideal unlearning. This provides an upper bound on achievable deletion performance and serves as a reference point for evaluating practical unlearning methods.
> >
> > > **Broader Impact**
> >
> > Our broader impact statement already mentions that “We view this work as a step toward safer medical AI systems, complementing, rather than replacing, other safeguards such as access control, auditability, and data minimization.”

---

### Review · Reviewer_SdPC · 2026-03-24

**Summary Of Contributions:**

The paper introduces MedForget, a benchmark for unlearning in medical MLLMs that organizes MIMIC-CXR data into a four-level hierarchy (institution, patient, study, section), and CHIP, a training-free unlearning method. CHIP computes "sibling-differential" directions (target activation mean minus sibling mean) and projects weights onto the orthogonal complement of that subspace. The idea is that subtracting sibling activations removes shared parent-level information, so the projection hits only what's specific to the forget target.

The hierarchy framing is well-motivated and the sibling-contrast idea is clean. The ablation in Table 8 shows convincingly that the sibling subtraction is what makes the method work. The writing is clear.

The main issues are that the headline claim is partly contradicted by the paper's own results table, all experiments are on a single LoRA-adapted model, there's no uncertainty reporting, and there are a few table/prose inconsistencies.

**Audience:**

Yes

**Audience Explanation:**

Machine unlearning for medical MLLMs is under-explored and has clear regulatory relevance. The hierarchy-aware framing and the sibling-differential idea are both things other researchers could build on.

**Broader Impact Concerns:**

The existing statement is fine. It could note more explicitly that all methods leave substantial residual memorization (forget Gen Scores around 39-45), so this shouldn't be read as regulatory-grade erasure.

**Claims And Evidence:**

No

**Claims Explanation:**

My concerns are about claim scope rather than the underlying work. I think most of this can be fixed by narrowing the claims, and I'd expect to change my answer after a revision along those lines.

The abstract says CHIP "achieves the highest forget-retain performance gap across all hierarchy levels." Table 2 shows that at the Section level, KL Min has a higher F/R Diff (9.99 vs. 6.56). This should say "three of four levels."

The F/R Diff metric also hides a mixed utility picture. CHIP has the lowest Retain Cloze Accuracy at every level (roughly 73-79, vs. ~99 for the training-based baselines), and it's near the bottom on General Med generation and cloze. It does best on General Med classification, so the picture isn't uniformly bad, but "competitive downstream utility" in the abstract doesn't reflect the cloze numbers.

The generality claims are too broad for what was tested. Everything is on LoRA-adapted Lingshu-7B with MIMIC-CXR. The institutions are random groupings of eight patients, so there's no real institutional covariate structure. "General solution" and "practical, HIPAA-aligned benchmark" should be scoped to match.

Two minor reporting issues: Table 2 and Table 7 disagree on NPO's Institution-level F/R Diff (7.99 vs. 9.99), and the A.8.2 prose gives Lang Only numbers that don't match Table 8.

**Requested Changes:**

Critical for acceptance:
1. Fix the "across all hierarchy levels" claim to match Table 2.
2. Acknowledge the cloze and general-med utility costs in the main text, not just the table.
3. Narrow the generality and realism claims to what was actually tested.
4. Fix the table/prose inconsistencies.

Would strengthen:
5. Some form of uncertainty estimates.
6. Note that DeepSeek-V3 both generates the data and judges the outputs.

---

> ### Author Response · Authors · 2026-04-13
> **Response to Reviewer SdPC: Part 1**
>
> We sincerely thank the reviewer for their careful and constructive feedback. They noted that **“the hierarchy framing is well-motivated and the sibling-contrast idea is clean,”** and highlighted that **“the ablation in Table 8 shows convincingly that the sibling subtraction is what makes the method work.”** They also appreciated that **“the writing is clear,”** and recognized that **machine unlearning for medical MLLMs is under-explored and has clear regulatory relevance**, with **the hierarchy-aware framing and sibling-differential idea being useful for future work**. We address each point raised by the reviewer below and highlight all corresponding revisions in the updated PDF in blue.
>
> > **Claim Scope and F/R Diff Interpretation.**
>
> We thank the reviewer for this careful reading and for pointing out the discrepancy. We agree that the original statement “across all hierarchy levels” is not supported by Table 2 in the initial submission. In the revised version, based on other reviewers’ suggestions, we expand the dataset to a larger and more diverse scale, and update all results accordingly. Under this updated setting,, we find that CHIP has the highest forget-retain (F/R) gap in all four hierarchy levels. We also agree that F/R Diff should not be interpreted in isolation, as it captures the forgetting–retention trade-off but does not fully reflect downstream utility. In the revised version, we have clarified this explicitly in Appendix A.12, noting that an ideal comparison would evaluate forget performance at matched retain performance levels across methods. However, achieving such alignment requires extensive hyperparameter tuning for each method and task, making it practically difficult to achieve. In this context, F/R Diff serves as a practical approximation for comparing the forgetting–retention trade-off.
>
> Regarding downstream utility, we acknowledge the reviewer’s observation about lower Retain Cloze Accuracy in the original submission. In the revised version, we have updated our evaluation (Table 2), where CHIP achieves retain cloze performance that is comparable to or higher than competing methods, while maintaining strong forgetting performance. We also moderate our claims around “competitive downstream utility” to better reflect the task-dependent nature of these results.
>
> > **Evaluation is restricted to a single MLLM**
>
> We extend our empirical evaluation to include an additional MLLM, MedGamma, alongside Lingshu-7B (Qwen2.5-VL). Across both architectures, we observe consistent trends: CHIP outperforms existing unlearning baselines on our benchmark, achieving stronger forgetting while maintaining competitive retain performance. These results suggest that the core components of CHIP, such as neuron selection and subspace projection, generalize beyond a single model family. Please see the new results in Appendix A.1, Table 3, and Table 9  in the revised paper.
>
> > **Dataset realism and structure.**
>
> We agree that the institution-level grouping in MedForget does not capture true institutional covariate structure, and we scope our claims accordingly in the revised version. In particular, we revise phrases such as “general solution” and “practical, HIPAA-aligned benchmark” to better reflect the intended scope.
>
> We clarify that MedForget is designed as a **controlled benchmark to study hierarchy-aware unlearning**, where the hierarchical structure (institution → patient → study → section) enables systematic evaluation of deletion behaviors across multiple granularities. The “HIPAA-aligned” aspect refers to **conceptual alignment with hierarchical deletion requirements** (e.g., removing all data associated with a patient or study), rather than claiming full realism of institutional variation.
>
> At the same time, the dataset is built on real clinical data (MIMIC-CXR), and in the revised version we expand it to include **larger scale, non-uniform hierarchical structures, and richer multi-level supervision**, making the evaluation setting more realistic and challenging. We include updated results on this expanded dataset (Sec 3.2, Table 2), which show consistent trends with our original findings.
>
> > **Minor reporting issue**
>
> Thanks for pointing this out. We have fixed this in Appendix A.9.2 in the revised paper and revised Tables 2 and 8 to remove the consistency.
>
> > **Some form of uncertainty estimates.**
>
> In the revised version, we incorporate uncertainty estimates by reporting mean and variance across 3 random seeds. This provides a clearer view of the stability of the results across runs. Please see Table 2 for the error bars.

---

> > ### Author Response · Authors · 2026-04-13
> > **Response to Reviewer SdPC: Part 2**
> >
> > > **DeepSeek-v3 used for both data generation and evaluation**
> >
> > In the revised version, we decouple generation and evaluation to mitigate this concern. Specifically, we use DeepSeek-V3 solely for synthetic data generation, and employ GPT-4o as an independent evaluator for computing the Factuality Score. This separation reduces potential circularity and evaluation bias. See Table 2 in the revised paper for the results trends that are consistent with previous evaluation based on DeepSeek-v3.
> >
> > > **Broader Impact**
> >
> > We explicitly add to the Broader Impact statement that all unlearning methods leave substantial residual memorization,  so this shouldn't be read as regulatory-grade erasure.

---

### Review · Reviewer_EgdB · 2026-03-29

**Summary Of Contributions:**

This paper introduces MedForget, a multimodal unlearning benchmark tailored for the medical domain. Unlike prior flat unlearning benchmarks, MedForget is uniquely structured around the inherent hierarchy of medical data. The benchmark includes 3840 question-answer pairs covering generation, cloze, and classification tasks based on MIMIC-CXR data. The authors show that existing unlearning methods struggle to achieve effective hierarchy-aware unlearning without degrading downstream medical utility on this benchmark.

To address this limitation, the paper proposes CHIP, a training-free unlearning method that computes sibling-differential directions to isolate target-specific information while preserving shared parent-level representations. By performing orthogonal weight projection on both the language backbone and the vision-language merger layers, CHIP effectively unlearns targeted information at various hierarchical levels. The authors conduct thorough experiments demonstrating that CHIP provides a superior deletion-utility trade-off and more robustness against hierarchical reconstruction attacks compared to baseline unlearning methods.

**Strengths:**
- The problem of hierarchical unlearning in medical MLLMs is well-motivated and relevant given privacy regulations like HIPAA.
- The proposed CHIP method is training-free, simple to implement, while showing good performance.
- The empirical evaluations are comprehensive, covering different hierarchical granularities and assessing vulnerability to hierarchical reconstruction attacks.

**Weaknesses:**
- The empirical evaluation is restricted to a single MLLM architecture (Lingshu-7B/Qwen2.5-VL), which limits the understanding of how CHIP (especially techniques such as neuro isolation, orthogonal projection) generalizes across different model families.
- The base model is fine-tuned exclusively with LoRA. It is unclear if different fine-tuning methods (e.g., full fine-tuning) would affect the effectiveness of the proposed unlearning method. LoRA limits learning to a low-rank subspace, which could potentially make neuron identification and projection methods more effective. Moreover, recent work has shown that LoRA inherently "learns less and forgets less" (Biderman et al., 2024), which might make unlearning somewhat easier in this setting compared to full fine-tuning.
- Factuality evaluation relies entirely on LLM-as-a-judge (DeepSeek-v3) for scoring.

**Audience:**

Yes

**Audience Explanation:**

The transition from flat to hierarchical unlearning addresses a existing gap in privacy-preserving AI. The proposed benchmark and method will be valuable to researchers and practitioners working on machine unlearning, multimodal learning, and medical AI.

**Claims And Evidence:**

Yes

**Claims Explanation:**

The paper backs its claims with a well-designed, comprehensive set of experiments. The MedForget benchmark maps to real clinical structures, and the authors systematically evaluate CHIP against various baselines. Thorough ablation studies justify the core components of the method, and the hierarchical reconstruction attacks demonstrate the model's robustness against information leakage.

**Requested Changes:**

**Critical**
- Could the authors please provide a discussion or preliminary results on the sensitivity of CHIP to the number of available sibling nodes. In real-world scenarios, a target node (e.g., a patient) might have very few or no non-target siblings in the retain set. How does this affect the calculation of the sibling-differential direction?

**Would simply strengthen the work:**
- Include an evaluation or discussion on how CHIP generalizes to other MLLM architectures or models fine-tuned with methods other than LoRA. Demonstrating success across different model families would underscore the general applicability of the approach.

---

> ### Author Response · Authors · 2026-04-13
> **Response to Reviewer EgdB**
>
> We sincerely thank the reviewer for their thoughtful and detailed evaluation. They noted that **“the problem of hierarchical unlearning in medical MLLMs is well-motivated and relevant given privacy regulations like HIPAA,”** and recognized that **MedForget is uniquely structured around the inherent hierarchy of medical data**. They also highlighted that **“the proposed CHIP method is training-free, simple to implement, while showing good performance,”** and appreciated that **“the empirical evaluations are comprehensive, covering different hierarchical granularities and assessing vulnerability to hierarchical reconstruction attacks.”** Additionally, they found that **“the paper backs its claims with a well-designed, comprehensive set of experiments,”** with **ablation studies and reconstruction attacks demonstrating robustness to information leakage**, and noted that the work **addresses a gap in privacy-preserving AI and will be valuable to researchers and practitioners**. We address each point raised by the reviewer below and highlight all corresponding revisions in the updated PDF in blue.
>
> > **Evaluation is restricted to a single MLLM**
>
> We extend our empirical evaluation to include an additional MLLM, MedGemma-4B, alongside Lingshu-7B (Qwen2.5-VL). Across both architectures, we observe consistent trends: CHIP outperforms existing unlearning baselines on our benchmark, achieving stronger forgetting while maintaining competitive retain performance. These results suggest that the core components of CHIP, such as neuron selection and subspace projection, generalize beyond a single model family. Please see the new results in Appendix A.1, Table 3, and Table 9 in the revised paper.
>
> > **Effect of Fine-Tuning Strategy on Unlearning Performance**
>
> We thank the reviewer for this observation. Importantly, all compared unlearning methods are evaluated under the same fine-tuning setting, ensuring that comparisons are fair and controlled. As a result, the relative performance differences, and the observed limitations of existing methods, are meaningful and informative. We expect these trends (e.g., hierarchy-dependent leakage and the privacy–utility trade-off) to generalize across different fine-tuning regimes, as they arise from the structure of the data and task rather than the specific parameterization of the model.
>
> We acknowledge that our evaluation is restricted to a specific fine-tuning regime. We clarify this explicitly in Appendix A.10 of the revised paper and scope our claims accordingly. Investigating how different fine-tuning strategies (e.g., full fine-tuning vs. parameter-efficient methods) interact with hierarchy-aware unlearning is an important direction for future work.
>
> > **DeepSeek-v3 for factuality evaluation**
>
> Following prior work \[1\] that adopts LLM-as-a-judge for evaluating generative outputs, we initially used DeepSeek-v3 for factuality assessment. In the revised version, we expand our evaluation to include an additional judge, GPT-5.4. We observe consistent trends across both judges: CHIP continues to outperform existing unlearning methods, achieving stronger forgetting while maintaining competitive retain performance. This consistency indicates that our conclusions are robust to the choice of evaluation model and are not an artifact of a specific LLM judge. Please note that the Gen Score we report is a weighted average of ROUGE (which is objective and not LLM-based) and an LLM-based factuality score.
>
> \[1\] Liu, Zheyuan, et al. "Modality-aware neuron pruning for unlearning in multimodal large language models." *Proceedings of the 63rd Annual Meeting of the Association for Computational Linguistics (Volume 1: Long Papers)*. 2025\.
>
> > **Sensitivity of CHIP to the number of available sibling nodes**
>
> When the number of available sibling nodes is very small, or in the extreme case where no non-target siblings are present in the retain set, the sibling-differential term reduces to a degenerate case. Specifically, in the absence of siblings, the sibling mean is undefined, and the direction effectively defaults to using only the target representation itself. In this setting, the method simplifies to removing the target-specific component without requiring contrast against siblings, and the rest of the pipeline (neuron selection, direction computation, and projection) remains unchanged.
>
> Importantly, this scenario does not introduce ambiguity in the objective: when no siblings exist, there is no risk of collateral damage to semantically adjacent nodes, and thus no need to preserve shared sibling information. As a result, the absence of siblings does not hinder the application of CHIP, but rather corresponds to a simpler unlearning setting where isolation of the target alone is sufficient. We discuss this in Appendix A.11 in the revised paper.

---

### Decision · Action_Editor_wZZT · 2026-06-02

**Recommendation:** Accept as is

**Additional Comments:**

N.A.

**Audience:**

Yes

**Audience Explanation:**

In the context of Large Language Models (LLMs), unlearning is known to be difficult. On the other hand, within the medical domain, the demand for effective unlearning technologies is particularly high for example, patient opt-out requests. This approach provides a solution to this challenge using the natural hierarchical structure in the medical domain, making it a relevant and compelling topic for researchers in this field.

**Claims And Evidence:**

Yes

**Claims Explanation:**

Existing multimodal question-answering (QA) unlearning benchmarks in the medical field suffer from a limitation that they fail to reflect the inherent hierarchical structure of the domain. To address this issue, this paper proposes a hierarchy-aware multimodal QA benchmark dataset for unlearning tasks in medical AI. Furthermore, it introduces CHIP, an unlearning method that leverages this hierarchical structure. The core claim of this paper is that the proposed method outperforms existing approaches while also achieving competitive performance on downstream tasks.

The reviewers pointed out that the evaluation of uncertainty and qualitative analysis are insufficient. While there is room for improvement,　the current evaluation provides reasonable support for the claims.

Regarding the first point (uncertainty), while evaluating across independent dataset partitions would yield a more robust assessment of uncertainty, the current approach already captures a baseline level of uncertainty.

Regarding the second point (qualitative analysis), although a more detailed qualitative evaluation would be desirable, the superiority of the proposed method is demonstrated through the quantitative results.

In conclusion, while addressing the reviewer's feedback would strengthen the paper, the current manuscript provides sufficient evidence to support its main conclusions.